# DEVIATION RATINGS: A GENERAL, CLONE INVARIANT RATING METHOD

## ABSTRACT

Many real-world multi-agent or multi-task evaluation scenarios can be naturally modelled as normal-form games due to inherent strategic (adversarial, cooperative, and mixed motive) interactions. These strategic interactions may be agentic (e.g. players trying to win), fundamental (e.g. cost vs quality), or complimentary (e.g. niche finding and specialization). In such a formulation, it is the strategies (actions, policies, agents, models, tasks, prompts, etc.) that are rated. However, the rating problem is complicated by redundancy and complexity of N-player strategic interactions. Repeated or similar strategies can distort ratings for those that counter or complement them. Previous work proposed "clone-invariant" ratings to handle such redundancies, but this was limited to two-player zero-sum (i.e. strictly competitive) interactions. This work introduces the first N-player general-sum clone-invariant rating, called *deviation ratings*, based on coarse correlated equilibria. The rating is explored on several domains including LLMs evaluation.

## 1 INTRODUCTION

Data often captures relationships within a set (e.g., chess match outcomes) or between sets (e.g., film ratings by demographics). These sets can represent anything including human players, machine learning models, tasks, or features. The interaction data, often scalar (win rates, scores, or other metrics), may be symmetric, asymmetric or arbitrary. These interactions can be strategic, either in an agentic sense (e.g., players aiming to win) or due to inherent trade-offs (e.g., cost vs quality). This can lead to a game-theoretic interpretation: sets as players, elements as strategies, and interaction statistics as payoffs. This framing is common in analyzing strategic interactions between entities like Premier League teams, chess players (Sanjaya et al., 2022), reinforcement learning agents and tasks (Balduzzi et al., 2018), or even language models (Chiang et al., 2024). More generally, the idea of formulating real-world interactions as normal-form games, empirical game-theoretic analysis (Wellman, 2006), is well explored.

The data obtained from such interactions are numerous so it is common to distill the performance of each strategy into a single scalar. Such a process is called a rating method. Many ratings have been proposed including Elo (Elo, 1978), Bradley-Terry (Bradley & Terry, 1952; Zermelo, 1929), Glicko (Glickman, 1995), TrueSkill (Herbrich et al., 2007), Nash averaging (Balduzzi et al., 2018), payoff rating (Marris et al., 2022), $\alpha$-Rank (Omidshafiei et al., 2019), and some based on social choice theory (Lanctot et al., 2024).

Although the real world is a complex multi-agent system, data evaluation rarely utilizes more than two players, nor non-zero-sum interactions. For instance, the leading language model leaderboard, Chatbot Arena (Chiang et al., 2024), is most naturally formulated as a three-player general-sum game (model vs. model vs. prompt). Where model players are competing to be best on each prompt, and a prompt player may favour difficult prompts, or prompts that can best distinguish between models. However, due to limitations of Elo, Chatbot Arena is only evaluated as a two-player zero-sum game (model vs. model) which overlooks interesting nuances of the rating problem, such as specialized models excelling on specific prompt subsets.

As well as being N-player general-sum, a rating method should also be *scalable*, *resilient*, and *directional*. Scalability, in the context of rating, is best motivated by Balduzzi et al. (2018). Performance on different tasks may measure identical skills. Over-representation of particular skills introduces biases into ratings. Such biases may only be detected post-hoc. Therefore, humans must curate

evaluation datasets to prevent redundant strategies skewing the results. Manual curation does not scale and a scalable rating method should "adjust automatically and gracefully to redundant data". For example, Chatbot Arena evaluates models on a wide variety of prompts which tests various underlying skills. Without curation or a scalable rating, the evaluation will be influenced by the distribution of prompts collected by users of Chatbot Arena, a population that may not represent the needs of all users. Resilience to manipulation is also important for rating schemes. In Chatbot Arena, any prompt can be submitted by anyone, any number of times. It would be possible to inflate a model's score by submitting many similar prompts that a model is known to excel at, even if the model is holistically not as strong. Finally a directional rating is one that is useful for driving continual improvement. Huge investment of resources is driven by companies hill-climbing on LLM leaderboards. It would be beneficial if hill-climbing ratings led to maximally improved models.

The three qualities discussed above can be achieved by a property known as "clone invariance"[1]: where copying strategies does not alter the ratings. The property is so important it has been continually rediscovered in multiple fields, notably by Nash averaging (Balduzzi et al., 2018), maximal lotteries (Kreweras, 1960; Fishburn, 1984; Brandt, 2017), and Yao's Principle (Yao, 1977), as well as others (Conitzer et al., 2024; Brandl et al., 2016; Laffond et al., 1993; Fisher & Ryan, 1995; Felsenthal & Machover, 1992; Rivest & Shen, 2010). Clone invariant methods are *scalable* because if one added redundant data it will not affect the rating. This allows evaluators to be *maximally inclusive*: all data can be included and no curation is necessary. Similarly, clone invariant methods are *resilient* to *clone attacks*: artificially skewed evaluation distributions will not change the resulting ratings. Clone invariant ratings offer better *directional* improvement. Hill-climbing on game theoretic evaluation methods has been shown to give better performance ([citation withheld to maintain anonymity], 2024).

All the clone-invariant methods are game-theoretic and involve computing a Nash equilibrium (NE) distribution. NE is convex and tractable to compute in two-player zero-sum games. However, in general NE is non-convex and intractable to compute in N-player general-sum games. In particular there are many disjoint equilibria, and it is not clear how to choose one (equilibrium selection problem (Harsanyi & Selten, 1988)). Unfortunately, for this reason, all current clone-invariant methods only work for two-player zero-sum games.

This work circumvents the problems with NE equilibrium selection by focusing on a related solution concept, coarse correlated equilibrium (CCE) (Hannan, 1957; Moulin & Vial, 1978). The result is the first N-player, general-sum, clone-invariant rating method: *deviation rating*. Deviation ratings are equilibrium based, and select for the strictest – most stable – equilibrium. Deviation ratings can be computed efficiently with linear programming. They are also unique, always exist, are offset invariant and dominance preserving. This work proves these properties hold for deviation ratings and the rating is explored in a number of contexts including the topical problem of LLM evaluation.

## 2 PRELIMINARIES

**Normal-Form Games**  Normal-form games (NFGs) model single timestep, simultaneous action strategic interactions between any number of $N$ players. Each player, $p \in [1, N]$, selects a strategy from a set $a_p \in \mathcal{A}_p = \{a_p^1, ..., a_p^{|\mathcal{A}_p|}\}$. A particular strategy is indexed by $a_p^i$, and $a_p$ is a variable that corresponds to a choice of strategy. A joint strategy contains a strategy for all players $a = (a_1, ..., a_N) \in \mathcal{A} = \otimes_p \mathcal{A}_p$, and is indexed $a^{ij\cdots}$. A payoff function $G_p : \mathcal{A} \mapsto \mathbb{R}$ maps a joint strategy to a payoff for each player. Most generally, this function can be a lookup table with $|\mathcal{A}|$ entries. Players may act stochastically, $\sigma_p \in \Delta^{|\mathcal{A}_p|-1} \; \forall p$, and in general may coordinate, $\sigma \in \Delta^{|\mathcal{A}|-1}$, where $\Delta$ is a probability simplex. Sometimes the notation $-p$ is used to mean "every player apart from $p$", for example $G_p(a) = G_p(a_1, ..., a_N) = G_p(a_p, a_{-p})$.

**Equilibria**  The expected deviation gain $\delta_p^\sigma : \mathcal{A}_p' \times \mathcal{A}_p'' \mapsto \mathbb{R}$ describes the expected change in payoff for a player $p$ when deviating to $a_p'$ from recommended action $a_p''$ under a joint distribution

---

[1]This property is also extensively studied in social choice theory, where it is known as the "independence of clones criterion" (Tideman, 1987).

$\sigma \in \Delta^{|\mathcal{A}|-1}$. This definition is related to regret.

$$\delta_p^\sigma(a_p', a_p'') = \sum_{a_{-p}} \sigma(a_p'', a_{-p}) \left[ G_p(a_p', a_{-p}) - G_p(a_p'', a_{-p}) \right] \tag{1}$$

The expected deviation gain directly relates to the definitions of approximate well-support correlated equilibria ($\epsilon$-WSCE) (Czumaj et al., 2014), approximate correlated equilibria ($\epsilon$-CE) (Aumann, 1974) and approximate coarse correlated equilibria ($\epsilon$-CCE) (Hannan, 1957; Moulin & Vial, 1978).

$$\epsilon\text{-WSCE:} \qquad \sigma \text{ s.t.} \qquad \delta_p^\sigma(a_p', a_p'') \leq \sigma_p(a_p'')\epsilon \qquad \forall p, a_p', a_p'' \tag{2a}$$

$$\epsilon\text{-CE:} \qquad \sigma \text{ s.t.} \qquad \delta_p^\sigma(a_p', a_p'') \leq \epsilon \qquad \forall p, a_p', a_p'' \tag{2b}$$

$$\epsilon\text{-CCE:} \qquad \sigma \text{ s.t.} \qquad \sum_{a_p''} \delta_p^\sigma(a_p', a_p'') \leq \epsilon \qquad \forall p, a_p' \tag{2c}$$

Every finite NFG has a nonempty set of (C)(WS)CEs. The set of $\epsilon$-(C)(WS)CEs is convex. Usually, parameter $\epsilon$ (the max-gain) is chosen to be 0, however when positive it defines an approximate equilibrium. For some games, feasible solutions exist for negative $\epsilon$ which correspond to strict equilibria. Nash equilibria (NE) can be defined using either Equation (2b) or Equation (2c), but have an additional constraint that in Equation (1), the joint must factorize, $\sigma(a) = \sigma_1(a_1)...\sigma_N(a_N)$. This is what makes NE, in general, non-convex. These solutions concepts are subsets of one another, WSNE $\subseteq$ NE $\subseteq$ WSCE $\subseteq$ CE $\subseteq$ CCE. This work focuses on CCEs, so we use simpler notation.

$$\text{CCE Deviation Gains:} \qquad \delta_p^\sigma(a_p') = \sum_{a_p''} \delta_p^\sigma(a_p', a_p'') = \sum_a \sigma(a) \left[ G_p(a_p', a_{-p}) - G_p(a) \right] \tag{3a}$$

$$\epsilon\text{-CCE:} \qquad \sigma \text{ s.t. } \delta_p^\sigma(a_p') \leq \epsilon \qquad \forall p, a_p' \tag{3b}$$

## 3 RATING DESIDERATA

A strategy rating, $r_p : \mathcal{A}_p \mapsto \mathbb{R}$, assigns a scalar to strategies. Similarly, a ranking, $r_p : \mathcal{A}_p \mapsto \mathbb{N}$, defines a (partial) ordering over strategies. Rankings can be inferred from ratings. Ratings attempt to summarize how good a strategy is in relation to the other available strategies, in the strategic context of an NFG.

### 3.1 DESIDERATA

There are several desiderata for formulating rating methods including tractability, permutation equivariance, and robustness. This section presents and extends important desiderata that are particularly important in *game theoretic* rating.

**Dominance Preserving**  If a strategy dominates another, $G_p(\tilde{a}_p, a_{-p}) \geq G_p(\hat{a}_p, a_{-p}) \, \forall a_{-p}$, then a dominance preserving rating should result in ratings, $r_p(\tilde{a}_p) \geq r_p(\hat{a}_p)$.

**Clone Invariance**  Consider adding an additional strategy to a game $\tilde{a}_p$, which is a copy of existing strategy such that $\tilde{G}_p(\tilde{a}_p, a_{-p}) = G_p(\hat{a}_p, a_{-p}) \, \forall a_{-p}$. A clone invariant rating would result in ratings $\tilde{r}_p(\tilde{a}_p) = \tilde{r}_p(\hat{a}_p) = r_p(\hat{a}_p)$ and $\tilde{r}_p(a_p) = r_p(a_p) \, \forall p, a_p$ (equal and unchanged from original ratings).

**Mixture Invariance**  Consider adding an additional strategy to a game $\tilde{a}_p$, which is a mixture of the existing strategies such that $\tilde{G}_p(\tilde{a}_p, a_{-p}) = \sum_{a_p} \tilde{\sigma}(a_p) G_p(a_p, a_{-p})$. A mixture invariant[2] rating would result in ratings $\tilde{r}_p(\tilde{a}_p) = \sum_{a_p} \tilde{\sigma}(a_p) r_p(a_p)$, and unchanged original ratings.

**Offset Invariance**  Consider a game $G_p \, \forall p \in [1, N]$, and another game $\tilde{G}_p \, \forall p \in [1, N]$, where $\tilde{G}_p(a_p, a_{-p}) = G_p(a) + b_p(a_{-p}) \, \forall p \in [1, N]$, and $b_p(a_{-p}) \in \mathbb{R} \, \forall a_{-p} \in \mathcal{A}_{-p}$ is an arbitrary offset. An offset invariant rating would have ratings $r_p(a_p) = \tilde{r}_p(a_p) \, \forall p \in [1, N], a_p \in \mathcal{A}_p$.

**Generality**  Some rating strategies are only defined for NFGs with particular structure in the game. This includes the number of players, if players are symmetric, or any restrictions on the payoff structure. General rating schemes will work for all NFGs: they are N-player general-sum.

---

[2]This is a novel term introduced in this work.

## 3.2 Rating Methods

**Uniform** The simplest way to rate strategies is to average over their payoffs, $r_p(a_p) = \frac{1}{|\mathcal{A}_{-p}|} \sum_{a_{-p}} G_p(a_p, a_{-p})$. The uniform rating is defined in general classes of games, is simple to compute, and is dominance preserving. However, it is not clone invariant nor offset invariant.

**Elo** Elo (Elo, 1978) is only defined for symmetric two-player zero-sum games. Elo is popular because one can infer the approximate win probability between two strategies by just comparing their relative ratings. It has a stochastic update rule and is widely using in sports ratings. However, it is not clone invariant nor offset invariant, and has a number of other well-documented drawbacks (Shah & Wainwright, 2018; Balduzzi et al., 2018; Bertrand et al., 2023; Lanctot et al., 2023).

**Nash average** An interesting game-theoretic rating, Nash averaging (Balduzzi et al., 2018), is only defined for two-player zero-sum games[3], $r_1(a_1) = \sum_{a_2} \sigma_2(a_2) G_1(a_1, a_2)$ and $r_2(a_2) = \sum_{a_1} \sigma_1(a_1) G_2(a_1, a_2)$ where $(\sigma_1(a_1), \sigma_2(a_2))$ is the maximum entropy Nash equilibrium. It is clone-invariant which gracefully handles rating in regimes with redundant data. The rating assigned to a strategy by Nash averaging is their expected payoff under this maximum entropy Nash equilibrium. This idea of a *payoff rating*, i.e. quantifying a strategy by its expected payoff against a Nash equilibrium, can be extended to other solutions concepts beyond two-player zero-sum games, such as CE and CCE (Marris et al., 2022). However, retaining the original invariance properties is non-trivial.

**Voting Methods** Another way to compare strategies is to rank them rather than rate them; one way to do so is using social choice theory (i.e. voting mechanisms). Voting-based evaluations have been used for multi-task benchmarks in NLP domains (Rofin et al., 2023) and for general agent evaluation (Lanctot et al., 2023). The main advantage of these methods is that they inherit certain robustness properties, such as clone-invariance (Fishburn, 1984). The main disadvantage is that the quantification of the strength of an assessment (comparison between strategies) is lost by construction due to ordinal outcomes.

**$\alpha$-Rank** One alternative to the payoff rating mentioned above is a *mass rating* (Marris et al., 2022), which corresponds to the probability mass of a strategy in an equilibrium (i.e. $r_p(a_p) = \sigma_p(a_p)$). One such mass rating scheme is $\alpha$-Rank (Omidshafiei et al., 2019). However, instead of using the mass of a Nash equilibrium, $\alpha$-Rank defines the rating of a strategy as its mass in the stationary distribution of a dynamical system between sets of pure strategies known as a Markov-Conley chain.

## 4 Deviation Rating

Typically, the approach for developing game theoretic rating algorithms is to find an equilibrium, and calculate a rating based on that equilibrium. This requires choosing a solution concept and uniquely selecting a single equilibrium from a set. This is not difficult, for example a maximum-entropy coarse correlated equilibrium (MECCE) satisfies these properties. However, if we wish the rating to be clone invariant, the equilibrium selection method needs to somehow be rating-consistent between a game and a larger game containing a clone. This property is hard to achieve for N-player general-sum games. For example, an MECCE would spread probability mass differently in the expanded game resulting in different ratings. An NE based rating, would have consistent ratings, provided one could reliably select for the same equilibrium each time. Chen et al. (2009) showed that NE problems do not admit an FPTAS unless PPAD $\subseteq$ P.

To overcome these problems we side-step selecting a rating-consistent equilibrium, and instead select for unique deviation gains, $\delta_p^\sigma(a_p')$ (Equation (3a)). We then define ratings from the deviation gains. We propose a game theoretic rating scheme based on CCEs.

$$\text{Deviation Rating:} \qquad r_p^{\text{CCE}}(a_p') = \delta_p^{\sigma^*}(a_p') = \sum_a \sigma^*(a) \left[ G_p(a_p', a_{-p}) - G_p(a) \right] \qquad (4)$$

---

[3]To extend to other game classes one would need a way to uniquely select a Nash equilibrium (the equilibrium selection problem (Harsanyi & Selten, 1988)). Marris et al. (2022) suggested using a limiting logit equilibrium (LLE) (McKelvey & Palfrey, 1995).

| | Dominant Pres. | Clone Inv. | Mixture Inv. | Offset Inv. | N-Player | General-Sum |
|---|---|---|---|---|---|---|
| Elo | ✓ | | | | | |
| Nash Averaging | ✓ | ✓ | ✓ | | | |
| Uniform Averaging | ✓ | | | | ✓ | ✓ |
| Payoff Rating | ✓ | | | | ✓ | ✓ |
| Deviation Rating | ✓ | ✓ | ✓ | ✓ | ✓ | ✓ |

Table 1: Comparison of rating methods.

**Algorithm 1** CCE Deviation Rating

1: $\hat{\mathcal{A}}_p \leftarrow \varnothing \ \ \forall p$
2: $r_p(a'_p) \leftarrow 0 \ \ \forall p, a'_p$
3: **while** $\hat{\mathcal{A}}_p \neq \mathcal{A}_p \ \ \forall p$ **do**
4: $\quad \min_{\sigma \in \Delta} \max_{p, a'_p \in \mathcal{A}_p \setminus \hat{\mathcal{A}}_p} \delta_p^\sigma(a'_p)$

$$\text{s.t. } \ \delta_p^\sigma(a'_p) = r_p(a'_p) \ \ \forall p, a'_p \in \hat{\mathcal{A}}_p \tag{5}$$

5: $\quad \bar{\mathcal{A}}_p \leftarrow$ active max constraints $\forall p$
6: $\quad r_p(a'_p) \leftarrow \delta_p^\sigma(a'_p) \ \ \forall p, a'_p \in \bar{\mathcal{A}}_p$
7: $\quad \hat{\mathcal{A}}_p \leftarrow \hat{\mathcal{A}}_p \cup \bar{\mathcal{A}}_p \ \ \forall p$
8: **end while**
9: **return** $r_p(a'_p) \ \ \forall p$

Note that it is possible for many equilibria, $\sigma^*$, to result in the same deviation gains, so we no longer have to uniquely select an equilibrium to calculate a unique rating.

### 4.1 ALGORITHM

This work's primary innovation is in how we select deviation gains in a way that preserves clone invariance. The two properties such a selection operator must have are: a) permutation equivariance and, b) clone invariance. The maximum and minimum functions are two functions with this property[4]. Maximizing the deviation gains is counter-intuitive because it does not result in equilibria, and if you limited the procedure to $\epsilon \leq 0$, it would likely find $r_p^{\text{CCE}}(a'_p) = 0 \ \forall p, a'_p$ because there are many more degrees of freedom in $\sigma$, than there are in the deviation gains. Therefore we opt to minimize the deviation gains (which is equivalent to finding the strictest equilibrium). Concretely, *iteratively* minimize the maximum deviation gain, freezing active constraints at each iteration (Algorithm 1).

Each iteration requires solving a linear programming (LP) (Murty, 1983) sub-problem. The inner max operator is implemented using a slack variable and inequality constraints. Each inequality constraint has an associated dual variable. Nonzero dual variables indicate that the constraint is active and can be frozen. There will always be at least one active constraint at optimum, therefore each iteration is guaranteed to freeze at least one more constraint. Therefore the algorithm requires at most $\sum_p |\mathcal{A}_p|$ outer iterations.

This process results in unique ratings and a possibly non-singleton set of CCE equilibria that all evaluate to the same rating. Because the ratings are calculated under an equilibrium, there are no strategies that a player has incentive to deviate to. The recursive procedure used to calculate the deviation ratings select the strictest possible equilibrium. Deviating from such an equilibrium will ensure loosing the maximum amount of payoff, and therefore this equilibrium is the most stable. Strict equilibria tend to have higher payoff, therefore the equilibrium selection criterion is a natural one, where strategies that can give high payoffs in practice are rated highly.

### 4.2 PROPERTIES

No general quantitative metrics exist for evaluating ratings. Inventing metrics that measure properties (e.g., some measure of clone-invariance) can be contrived and circular. Therefore the literature tends to favour a qualitative approach, where properties are enumerated and proven. This section follows this approach. Comparisons to other ratings are found in Table 1.

**Property 1** (Existence). *Deviation ratings always exist.*

*Proof.* Deviation ratings are calculated from CCEs, a superset of NEs, which are known to always exist for finite normal-form games (Nash, 1951). □

**Property 2** (Uniqueness). *Deviation ratings are unique.*

---

[4]We are unaware of any other nontrivial operators with these properties.

*Proof.* The problem (Equation (5)) is convex, so the optimal objective is unique. The rating is derived from the objective value, not the (possibly non-unique) parameters, therefore the rating is unique. $\square$

**Property 3** (Bounds). *Deviation ratings are bounded:* $\min_a \left[ G_p(a'_p, a_{-p}) - G_p(a) \right] \leq r_p(a'_p) \leq 0.$

*Proof.* CCEs with $\epsilon = 0$ always exist. Therefore the maximum possible expected deviation rating is 0 and $r_p(a'_p) \leq 0 \; \forall p, a'_p$. The lower bound follows from the definition. $\square$

**Property 4** (Dominance Preserving). *Deviation ratings are dominance preserving.*

*Proof.* When $G_p(\tilde{a}'_p, a_{-p}) \geq G_p(\hat{a}'_p, a_{-p}) \; \forall a_{-p} \in \mathcal{A}_{-p}$, it follows that $G_p(\tilde{a}'_p, a_{-p}) - G_p(a) \geq G_p(\hat{a}'_p, a_{-p}) - G_p(a) \; \forall a \in \mathcal{A}$. Therefore, for any distribution $\sigma$, $\delta_p^\sigma(\tilde{a}'_p) \geq \delta_p^\sigma(\hat{a}'_p)$ and hence $r_p(\tilde{a}'_p) \geq r_p(\hat{a}'_p)$. $\square$

**Property 5** (Offset Invariant). *Deviation ratings are offset invariant.*

*Proof.* Consider a modified game with an offset $\tilde{G}_p(a) = G_p(a) + b_p(a_{-p})$. It is known that such an offset does not change the deviation gains (Marris et al., 2023): $\tilde{G}_p(a'_p, a_{-p}) - \tilde{G}_p(a) = G_p(a'_p, a_{-p}) - G_p(a)$, nor the set of equilibria. Therefore $\tilde{r}_p(a'_p) = r_p(a'_p) \; \forall p, a'_p$. $\square$

**Property 6** (Clone Invariant). *Deviation ratings are clone invariant.*

*Proof.* CCE (Equation (3b)) are defined by linear inequality constraints, $A\sigma \leq 0$, where $A$ is a constraint matrix with shape $[\sum_p |\mathcal{A}_p|, |\mathcal{A}|]$ and $\sigma$ is a flat joint distribution column vector with shape $[|\mathcal{A}|]$.

An additional strategy adds 1 row and $|\mathcal{A}_{-p}|$ columns to $A$, and $|\mathcal{A}_{-p}|$ rows to $\sigma$, therefore increasing the dimensionality. For example, when cloning strategy $a_p^i$, the resulting constraint matrix will have a transformed structure (after permuting rows and columns for clarity, and using Numpy indexing style notation):

$$A = \begin{bmatrix} A[\neg a_p^i, :] \\ A[a_p^i, :] \end{bmatrix} \quad \hat{A} = \begin{bmatrix} A[\neg a_p^i, :] & A[\neg a_p^i, \text{if } \hat{a}_p^i \in a] \\ A[a_p^i, :] & 0 \\ A[a_p^i, :] & 0 \end{bmatrix}. \tag{6}$$

The new strategy results in an identical row in the constraint matrix and is therefore redundant and can be ignored. The additional columns are copies of other columns. Therefore every equilibria in the un-cloned game has a continuum of equilibria in the cloned game corresponding to mixtures over the cloned actions. Importantly, the increased space of equilibria do not change the values the deviation gains can take. Therefore any method that uniquely selects over deviation gains will be clone invariant. $\square$

**Property 7** (Mixture Invariant). *Deviation ratings are mixture invariant.*

*Proof.* An additional mixed strategy results in an additional mixed constraint. This constraint is redundant, and any distribution will have an expected deviation gain which is the same mixture over other actions deviation gains. $\square$

**Property 8** (NA Special Case). *In two-player zero-sum games, Deviation ratings are a generalization of Nash averaging up to a constant offset* $r_p^{CCE}(a'_p) = r_p^{NA}(a'_p) - \sum_a \sigma(a) G_p(a)$.

*Proof.* The set of NEs, and CCEs is equal in nontrivial two-player zero-sum games and all equilibria in two-player zero-sum games have equal value, therefore differences in the equilibrium selection method unimportant.

$$r_p^{\text{NA}}(a'_p) = \prod_{-p} \sigma_p(a_p) G_p(a'_p, a_{-p}) = \sum_a \sigma(a) G_p(a'_p, a_{-p}) = r_p^{\text{CCE}}(a'_p) + \sum_a \sigma(a) G_p(a)$$

$\square$

|   | R | P | S | N |
|---|---|---|---|---|
| R | $-8,-8$ | $-2,+2$ | $+4,-4$ | $\frac{-680}{241},\frac{-712}{241}$ |
| P | $+2,-2$ | $-8,-8$ | $-1,+1$ | $\frac{-680}{241},\frac{-920}{241}$ |
| S | $-4,+4$ | $+1,-1$ | $-8,-8$ | $\frac{-680}{241},\frac{-184}{241}$ |
| N | $\frac{-712}{241},\frac{-680}{241}$ | $\frac{-920}{241},\frac{-680}{241}$ | $\frac{-184}{241},\frac{-680}{241}$ | $\frac{-680}{241},\frac{-680}{241}$ |

(a) Biased Shapley Payoffs

|   | $r^{\text{Uni}}$ | $r^{\text{CCE}}$ |
|---|---|---|
| R | $\frac{-2126}{964}$ | $\frac{-2720}{964}$ |
| P | $\frac{-2367}{964}$ | $\frac{-2720}{964}$ |
| S | $\frac{-3331}{964}$ | $\frac{-2720}{964}$ |
| N | $\frac{-2497}{964}$ | $\frac{-2720}{964}$ |

(b) Ratings

Table 2: The payoffs (a) and ratings (b) of a biased Shapley's game with an augmented Nash strategy. The game contains a cycle, R $\succ$ S $\succ$ P $\succ$ R $\succ$ ..., and penalises when both players player the same strategy.

## 5 ILLUSTRATIVE STUDIES

The qualitative properties used to motive deviation rating have been proven, but their usefulness may not yet be apparent. Therefore this section is intended to build intuition, highlight the properties of deviation ratings, and demonstrate the diversity of applications.

### 5.1 RATINGS IN CYCLIC AND COORDINATION ENVIRONMENTS

Shapley's game (Shapley, 1964) (Shoham & Leyton-Brown, 2009, p210) is a symmetric general-sum variant of rock-paper-scissors with losing payoffs for each player if they play the same strategy. Therefore it is a cyclic anti-coordination game. In the unbiased form of the game, there is a single mixed Nash equilibrium, $\left[\frac{1}{3}, \frac{1}{3}, \frac{1}{3}\right]$. We consider a biased version of such a game (Table 2a) with a single mixed Nash equilibrium $\left[\frac{87}{241}, \frac{100}{241}, \frac{54}{241}\right]$.

A uniform rating of the strategies produces a transitive ranking R $\succ$ P $\succ$ N $\succ$ S (Table 2b). This is because, ignoring strategic interactions, the biases separate the strategies. For example, rock is particularly effective at defeating scissors, and all possible opponents are considered equally when using uniform rating. This is unrealistic because if scissors is vulnerable, one may expect to encounter that strategy less frequently and therefore perhaps less attention should be placed on strategies that defeat it. Furthermore, the uniform rating scheme ranks the Nash strategy second last. This is unfortunate because the Nash strategy is the only unexploitable pure strategy in this game, and arguably should be ranked the highest. In contrast, the deviation rating result in equal ratings R = P = S = N. From a game theoretic perspective, this makes intuitive sense: while rock, paper, and scissors all appear in a cycle, and dominate each other, no strategy can be said to be better than another. Similarly, the Nash strategy is a special mixture of the others such that it has the same expected payoff, therefore it should also be rated equally.

Now let us sample mixed policies from the biased Shapley game to produce a population of strategies, resulting in an expanded symmetric NFG with number of strategies equal to the number of samples. Each strategy is a mixture of the "pure" strategies: R, P, and S. We analyse the ratings of strategies in populations drawn from different distributions to observe how the distribution affects the ratings.

Firstly, consider unbiased sampling (Figure 1a). The uniform rating still rates rock the highest. The other strategies in the population are rated linearly across the space with R $\succ$ P $\succ$ S. Deviation ratings continue to rank all strategies equally (due to mixture invariance). Interestingly, equilibrium mass is placed only on the convex hull of the population. Now, consider a biased population where most mixtures play close to paper (Figure 1b). The uniform rating now favours scissors which counters paper: S $\succ$ R $\succ$ P. However, deviation rating continues to rate all strategies equally. It is clear that by manipulating the distribution, the uniform rating can be made to rate any of R, P or S the highest. While the deviation rating will always rate them equally.

Slightly restricting the domain of the population (Figure 1c), means there is still a cycle, also does not affect the ratings. A population with only minority scissor players (Figure 1d) should favour paper. There is no longer a cycle, and in a world of rock and paper, paper is king. However there is still an anti-coordination aspect to the game which is why both R and P get probability mass under the equilibrium. The uniform rating rates the most mixed strategy the highest because it is best at avoiding coordination across the distribution.

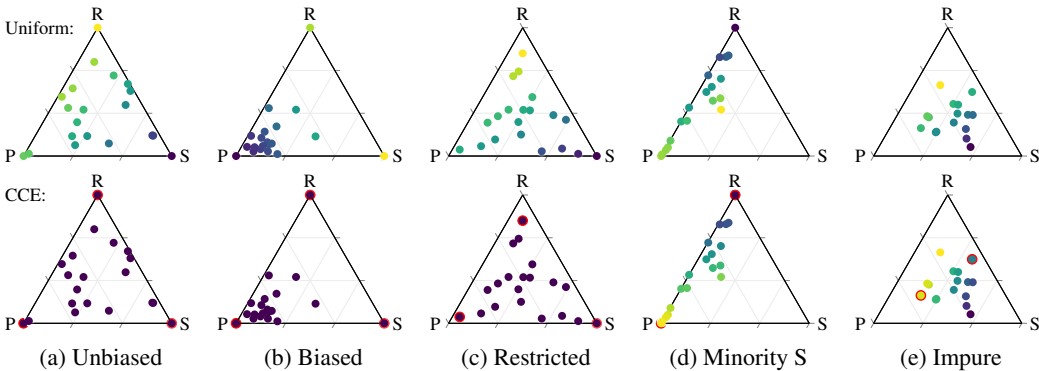

(a) Unbiased     (b) Biased     (c) Restricted     (d) Minority S     (e) Impure

Figure 1: Population ratings for Shapley's game. The position of the points indicates the underlying mixture of each strategy. The fill color of the point represents its rating under the rating function. The outline color represents the marginal probability mass each strategy has under the equilibrium. Each column is a different population distribution. Top: Uniform, bottom: CCE.

Sampling a population without having the pure strategies in the convex hull of the population (Figure 1e) results in in an NFG which no longer has three underlying strategies that the others are mixtures of. It instead has the number equal to the convex hull of the population. This game looks like an anti-coordination game and, the population is rated as such.

## 5.2 LANGUAGE MODEL RATING

There are many leaderboards for evaluating LLMs including Chatbot Arena (Chiang et al., 2024), where language models are evaluated on pairwise matchups on a prompt. The model that gives the better response wins. The final ratings are aggregate Elo ratings over many prompts. The ratings are published as a popular and trusted leaderboard of LLMs. However, the Elo ratings depend on the distribution of prompts that are submitted. Therefore popular prompts will drive the ratings. People who submit prompts to Chatbot Arena may not be representative of end users of LLMs nor the tasks they wish to perform with them. Companies developing LLMs may miss important functionalities if they optimize only for such benchmarks.

A more game theoretic approach would be to evaluate the models in the context of a three player game: prompt vs model vs model. The model players' payoffs are symmetric zero-sum evaluations over every prompt. The prompt player's payoff is the maximum of the two model players: $G_P(a) = \text{maximum}[G_{M_A}(a), G_{M_B}(a)] = |G_{M_A}(a)| = |G_{M_B}(a)|$. Therefore the prompt player either has a zero-sum or common-payoff interaction with each model player, depending on who is winning the prompt, and favours selecting prompts that separate the models.

Because Chatbot Arena only has comparison data between two models for each prompt, and we require all models to be evaluated, we instead focus on another benchmark: Livebench (White et al., 2024). Livebench evaluates language models across 18 tasks (curated sets of prompts) resulting in a model vs task dataset. Evaluating models against tasks using the methodology discussed in Balduzzi et al. (2018) is unsatisfying (Lanctot et al., 2023) because models are adversarially evaluated against the hardest tasks.

Our actual objective is to evaluate models relative to other models, in the context of tasks. Therefore, from the model vs task data $T(m,t)$ (Figure 2c) [5], let us construct a three player model vs model vs task game with payoffs: $G_A(m_A, m_B, t) = T(m_A, t) - T(m_B, t), G_B = -G_A, G_T = |G_A| = |G_B|$. This is similar to the Chatbot Arena game formulation but is derived from only model vs task data.

Uniform and Elo in this game result in close to identical ratings (Figure 2a). The deviation ratings place four models equally at the top: `claude-3-5-sonnet`, `gemini-1.5-pro`, `Llama-3.1-405B` and `gpt-4o`. The grouping property is typical of game theoretic solvers

---

[5]`https://huggingface.co/datasets/livebench/model_judgment` (2024/08/18)

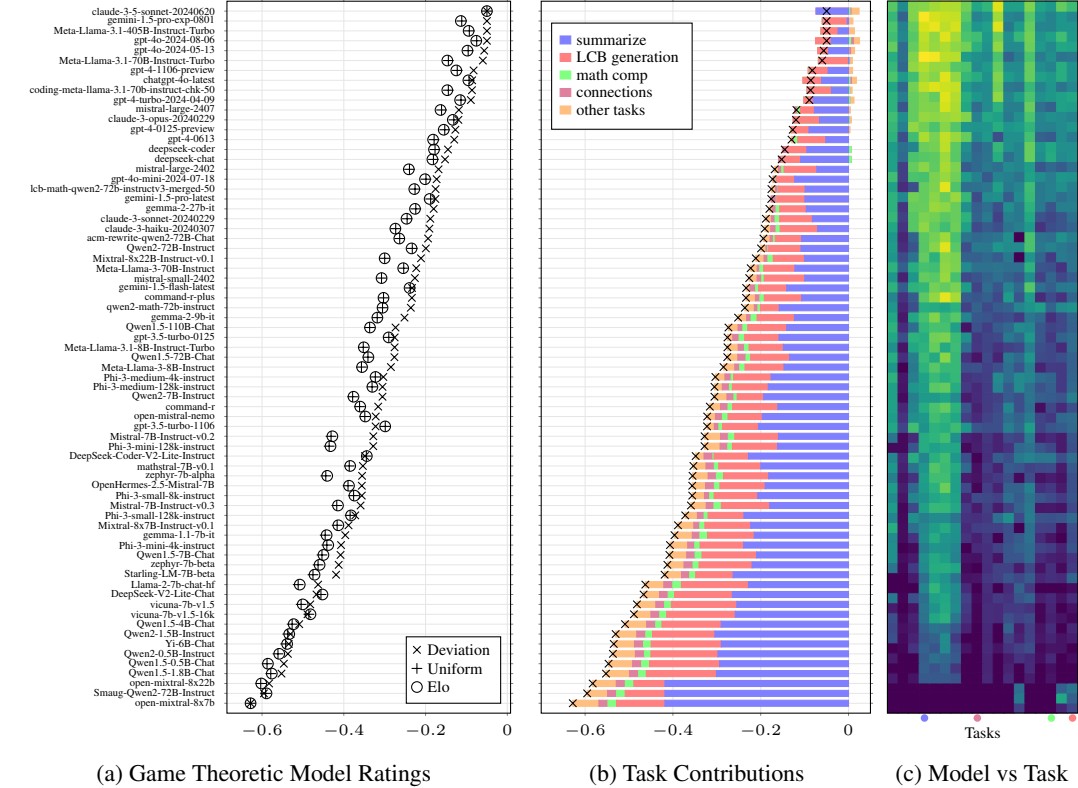

(a) Game Theoretic Model Ratings     (b) Task Contributions     (c) Model vs Task

Figure 2: Livebench analysis. (a) Model ratings with competing evaluation algorithms. Uniform and Elo ratings have been rescaled to fit into the same domain as the CCE deviation ratings. (b) Analysis showing how the four most salient tasks contributes to the CCE deviation rating. The bars sum to the corresponding ratings. (c) The full raw model vs task data used for evaluation.

and arises because models are better than others at certain tasks. We can analyse task contributions (Figure 2b) by examining how the rating will change when deviating from the CCE distribution, segregated over each task. Concretely, by computing $c(m'_A, t) = \sum_{m_A, m_B} \sigma^*(m'_A, m_B, t)[G_A(m'_A, m_B, t) - G_A(m_A, m_B, t)]$. Note that these statistics relate to the ratings themselves $r_A(m'_A) = \sum_t c(m'_A, t)$. For example, `claude-3-5-sonnet` is good at LCB generation, `gemini-1.5-pro` is good at summarize, `Llama-3.1-405B` is good at other, and `gpt-4o` is good at connections. The rating scheme emphasises tasks that are particularly good at separating the top models, so it also serves as an important tool when developing evaluation datasets.

The deviation ratings seem to capture an intuition that people have when interpreting evaluation data: there are different competency measures and if no one solution is best then it is fraught to separate solutions that fill the different niches without further assumptions. It is best to group the strong models together and say that each has its relative strengths and weaknesses. Or course, if one model was truly dominant across all tasks, the deviation rating would rate it the highest, because deviation rating is dominance preserving.

## 5.3 RATINGS TO DRIVE MODEL IMPROVEMENT

One main use of ratings is to drive improvement of models. Fair and representative ratings inform how companies fund, develop, train, and improve upon existing models. Because resources are often constrained, only a handful of alternative models can be maintained. This small population of models has to suffice to properly evaluate changes and ensure that progress is being made. We simulate such a development process by searching for policies that could represent an equilibrium in extensive-form games. Games have interesting structure, strategic trade-offs, and necessitate

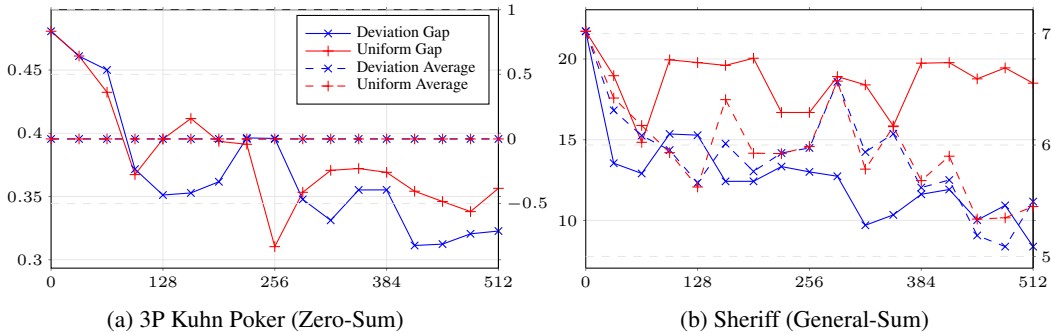

(a) 3P Kuhn Poker (Zero-Sum)  (b) Sheriff (General-Sum)

Figure 3: Model improvement analysis. Shows the equilibrium gap (left axis, lower better) and average payoffs (right axis, higher better) with iteration count over two OpenSpiel (Lanctot et al., 2019) environments.

maintaining diverse tactics, which make them suitable environments to study. However, extensive-form games grow exponentially in size as a function of the action sequence length; solving them empirically through simulation has emerged as a natural approximation technique (Wellman, 2006).

The simulation is initialized with a population of 8 randomly sampled stochastic policies for each player and then follows a loop: a) construct a meta-game which describes the payoffs between policies, b) rate the policies, c) discard the bottom quarter, d) replace bottom quarter with new random policies.

To measure progress, at each iteration we compute the analytical distance to equilibrium (i.e. CCE gap, $\sum_p \max_{a'_p} \delta^\sigma_p(a'_p)$), Equation (3a)) in the *full game*, by traversing the game-tree, from a distribution[6] over the policies in the population. The CCE gap over the full game gives a more holistic summary of the strength of the population than the myopic ratings over the meta-game could achieve. The thesis is that game-theoretic meta-game ratings are better equipped at selecting policies for equilibrium representation in the overall landscape of the game, despite limited samples. Therefore, in the simple evolutionary loop described above, we expect that deviation ratings should be better fitness measures for the population policies. Additionally we track the average payoff for the policies in the population.

We find (Figure 3) that both uniform and deviation ratings can drive a reduction in the gap in a zero-sum game. However, in a general-sum game, deviation gain is only able to drive a reduction in the gap in a general-sum game. The average payoff reduces about similarly for both rating methods. Theory does not predict that this should necessarily increase in the setting we are studying. Seemingly high average payoff strategies may be exploited.

## 6 CONCLUSION

This work introduces deviation rating, a novel rating algorithm that produces unique, dominance preserving, clone invariant, mixture invariant, and offset invariant ratings for the most general class of N-player general-sum normal-form games. The method is the first clone-invariant rating algorithm for N-player general-sum games. Ratings can be formulated as sequential linear programs, and therefore many off-the-shelf solvers can compute the ratings in polynomial time. Such a rating scheme allows for scalable, maximally inclusive, clone-attack-proof, data agnostic rating as it naturally weights strategies according to their relevance in a strategic interaction. Clones and mixtures do not affect ratings at all. The rating is applicable in general strategic interactions and we highlight its utility in rating LLMs.

---

[6]Any selection criterion will do, we use maximum entropy (MECCE) (Ortiz et al., 2007).

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

## A BACKGROUND ON RATING METHODS

### A.1 ELO

Elo (Elo, 1978) is the most well-known rating system for rating players in *symmetric* two-player zero-sum games, and is most famous for its use in Chess. Its main feature is that the ratings of two players can be used to predict the win probability. The probability player $A$ beats player $B$ can be calculated $\hat{p}_{AB} = (1 + 10^{400r_B - 400r_A})^{-1}$, where 400 and 10 are arbitrary constants that scale the ratings. The main advantage of Elo is that it can be updated using sparse samples of player interactions: it does not require knowing the dense win-probabilities between all players.

If one did know the win probabilities, $p_{ij}$, Elo can be formulated as the solution, $\min_r L$, to a logistic regression problem. This form is sometimes known as BayesElo.

$$L = \sum_{ij} - p_{ij} \log_{10}(\hat{p}_{ij}) - (1 - p_{ij}) \log_{10}(1 - \hat{p}_{ij}) \tag{7}$$

Elo has a number of well-known drawbacks (Shah & Wainwright, 2018; Balduzzi et al., 2018; Bertrand et al., 2023; Lanctot et al., 2023). Most importantly, it only works for symmetric two-player games with probability payoffs, it is not clone-invariant, and can suffer large error in predictions.

Like most other rating methods, Elo cannot account for intransitive payoffs (e.g., rock-paper-scissor-like cycles), because a player's strength has to be summarized by a single number. By allowing multiple parameters per-player, it may be possible to capture cycles in the data. Multidimensional Elo (Balduzzi et al., 2018) is an extension that has this property.

Despite its drawbacks Elo remains popular in machine learning. It is used as the rating method in Chatbot Arena (Chiang et al., 2024), and is used to measure the performance of game-playing agents (Silver et al., 2016; Vinyals et al., 2019).

### A.2 SOCIAL CHOICE

Social choice theory focuses study on voting systems. There are a number of desirable criteria voting systems can hold. Two commonly studied criteria are the "independence of clones criterion" (Tideman, 1987) and "independence of irrelevant alternatives" (IIA) (Shoham & Leyton-Brown, 2009, p261). Independence of clones is the same as clone-invariance discussed in the main text. It concerned with the problem of similar candidates splitting the vote and changing election results. If a voting system has the IIA property, it means that the position of two agents $\{A, B\}$ in the ranking depends only the relative ordering of $\{A, B\}$ in the voters' preferences, not *e.g.,* on some other irrelevant agent $C$.

An impossibility result, Arrow's impossibility theorem (Arrow, 1950), stated that there is no deterministic rating rule that satisfies both independence of clones and independence of irrelevant alternatives. However it was shown that one could get around this by casting voting as a two-player zero-sum game, using probabilistic choices, and finding a minimax solution (von Neumann & Morgenstern, 1947). The payoffs of the game are integers, $G_p(a) \in \mathbb{Z} \, \forall a \in \mathcal{A}$, where each element corresponding to a comparison between two alternatives, say $A$ and $B$. The payoff is defined as the number of voters that prefer $a$ to $b$ minus the number of voters that prefer $A$ to $B$. The resulting method, maximal lotteries (Kreweras, 1965; Fishburn, 1984; Brandt, 2017), was the first method shown to satisfy both independence of clones and independence of irrelevant alternatives. Maximal lotteries is so fundamental it has been rediscovered multiple times (Conitzer et al., 2024; Brandl et al., 2016; Laffond et al., 1993; Fisher & Ryan, 1995; Felsenthal & Machover, 1992; Rivest & Shen, 2010).

Nash averaging (Balduzzi et al., 2018) is similar to maximal lotteries, but Nash averaging allows for arbitrary two-player zero-sum payoffs, $G_p(a) \in \mathbb{R} \, \forall a \in \mathcal{A}$. For example, in the original paper, a game is derived directly from agent scores in the Atari Learning Environment Bellemare et al. (2013). Solving for a Nash equilibrium is equivalent to a minimax solution in two-player zero-sum games.

### A.3 EQUILIBRIUM-BASED RATINGS

Nash averaging (Balduzzi et al., 2018) rediscovered maximal lotteries but with a notable generalization: it uses arbitrary two-player zero-sum payoffs. Balduzzi et al. (2018) motivated the rating in the context of evaluating general machine learning models, where models are required to perform well on a variety of different tasks, with each task testing potentially overlapping skills. Although the work focused on reinforcement learning agents, the problem is now also important for language models which are required to respond correctly to a wide variety of prompts.

A key property of Nash averaging is that it is clone-invariant. This means that repeated strategies in the game do not affect the ratings of the other strategies, which is important in the context of evaluating agents against tasks which measure overlapping skills. This allows the rating problem to be maximally inclusive with evaluation data, avoids the need for curation, and is therefore scalable.

Work to extend Nash averaging, such as payoff ratings (Marris et al., 2022), utilizes equilibria to rate strategies in N-player general-sum games. However, clone-invariance is not guaranteed.

### A.4 EVOLUTIONARY-BASED RATINGS

Another game-theoretic rating method, $\alpha$-Rank (Omidshafiei et al., 2019), is the stationary distribution of a dynamical system between sets of pure strategies know as a Markov-Conley chain. The ratings are the resulting mass distributions over strategies. Strategies with more mass are rated higher.

### A.5 OTHER METHODS

There are many other ratings methods, including TrueSkill (Herbrich et al., 2007), Glicko (Glickman, 1995), and voting-as-evaluation (Lanctot et al., 2023).

## B PRACTICAL COMPUTATION

Algorithm 1 sequentially solves linear programs (LPs). In the worst case, deviation ratings require $\sum_p |\mathcal{A}_p|$ outer iterations (the number of constraints in the deviation gains). The LP inner loop can be solved using many algorithms (simplex (Dantzig, 1956), ellipsoid (Khachiyan, 1979)) for which there are many off-the-shelf solvers (GLOP (Perron & Furnon), Gurobi (Gurobi Optimization, LLC, 2024), ECOS (Domahidi et al., 2013), OSQP (Stellato et al., 2020)) and many frameworks (CVXPY (Diamond & Boyd, 2016; Agrawal et al., 2018)). LPs can be solved in polynomial time (Khachiyan, 1979). Therefore deviation ratings can also be solved in polynomial time. Because the algorithm solves a similar problem multiple times it is advantageous to leverage disciplined parameterized programming (DPP) (Agrawal et al., 2019) to eliminate the need to recompile the problem at each outer iteration. Additionally, because the problem is solved repeatedly, care needs to be taken to minimize the accumulation of errors.

### B.1 SYMMETRIES

Exploit all symmetries in the problem to improve conditioning, and reduce solve time. There are three main symmetries that can be removed: payoff symmetries, joint symmetries, and constraint/strategy symmetries. These symmetries are best dealt with by manipulating the constraint matrix, $A$, with shape $[C, |A_1|, ..., |A_N|]$.

**Payoff Symmetries** Frequently, the payoffs may be symmetric across two players by construction (for example in model vs model). Incorporating this information has two benefits. Firstly, it reduces the number of variables to optimize over by half. Secondly, it makes the optimization problem less ill-conditioned. For example, the simplex algorithm may suffer from "small pivots" if payoff symmetries are not removed.

To remove payoff symmetries modify the constraints payoff by averaging over the symmetry permutations. For example, in a two player symmetry across players $p$ and $q$:

$$A[c, ..., a_p, ..., a_q, ...] = \tfrac{1}{2} \left( A[c, ..., a_p, ..., a_q, ...] + A[c, ..., a_q, ..., a_p, ...] \right) \qquad (8)$$

This will result in a constraint matrix, when viewed flat, $A[c, a]$, with repeated columns. These repeated columns can be pruned (see joint symmetries below).

Doing this preprocessing step will mean that only symmetric equilibria can be found. This is ideal for our purposes and will not alter any rating values.

**Joint Symmetries**    Columns in the constraint matrix (which correspond to joint strategies) may be repeated. This can occur if there are payoff symmetries, repeated strategies, or because of naturally occurring structure. Under the objectives we optimize for, probability mass can be arbitrarily mixed between repeated joint strategies without changing the deviation gains. Therefore we only need to track one of these joints. Counts should be tracked, to a final full dimensional joint can be reconstructed after a solution has been found.

### B.2    Quantization

Some solvers may struggle with differences close to numerical precision. We find that quantizing to 14 decimal places is sufficient to eliminate ill-conditioning caused by this problem. Such small quantization has negligible effects on the ratings.

### B.3    Algorithm Implementation

For the algorithm implementation in this paper we used CVXPY (Diamond & Boyd, 2016; Agrawal et al., 2018) with GLOP (Perron & Furnon) as the solver backend. GLOP is a free (available in OR-Tools[7]), single-threaded, primal-dual simplex, linear programming solver. We used default GLOP parameters[8] and ran the experiments on consumer-grade CPU hardware.

## C    Evaluation Studies

### C.1    Ratings to Drive Model Improvement

We used extensive-form environments from OpenSpiel (Lanctot et al., 2019). The library also includes code for sampling random policies, calculating expected returns, and calculating CCE gap.

**Kuhn Poker**    Kuhn poker (Kuhn, 1950) is a very simple zero-sum poker variant, with only up to two actions at each infostate (bet and pass). We use a three player variant of the game.

**Sheriff**    Sheriff (Farina et al., 2019) is a general-sum negotiation game. Parameters: item penalty 1, item value 5, max bribe 2, max items 10, number of rounds 2, and sheriff penalty 1.

## D    Further Evaluation Studies

### D.1    Atari Agents

We amalgamated (Table 3) reinforcement learning agent evaluation data on the Atari learning environment (Bellemare et al., 2013) sourced from numerous papers (Figure 4a).

We rated (Figure 4b) the agents using uniform and deviation ratings in two gamification regimes. Firstly, the agent vs task regime, motivated by Balduzzi et al. (2018). This regime normalizes the evaluation data across the game dimension so that each game has similar payoff ranges and constructs a two player zero-sum game with the agent player maximizing the payoff and the task player minimizing it. This creates an adversarial setting where the agents are primarily rated on the hardest tasks. Secondly, we rate in the agent vs agent vs task regime motivated in this paper. This approach is a three-player general sum game, with zero-sum interactions between the agents, and

---

[7]https://github.com/google/or-tools
[8]https://github.com/google/or-tools/blob/stable/ortools/glop/parameters.proto

| Agent | Agent Reference | Data Reference |
|---|---|---|
| r2d2(bandit) | (Kapturowski et al., 2019) | (Badia et al., 2020a, Sec H.4) |
| agent57 | (Badia et al., 2020a) | (Badia et al., 2020a, Sec H.4) |
| muzero | (Schrittwieser et al., 2019) | (Badia et al., 2020a, Sec H.4) |
| r2d2 | (Kapturowski et al., 2019) | (Badia et al., 2020a, Sec H.4) |
| r2d2(retrace) | (Kapturowski et al., 2019) | (Badia et al., 2020a, Sec H.4) |
| ngu | (Badia et al., 2020b) | (Badia et al., 2020a, Sec H.4) |
| muesli | (Hessel et al., 2022) | (Hessel et al., 2022, Tab 11) |
| muzero2 | | (Hessel et al., 2022, Tab 11) |
| rainbow | (Hessel et al., 2017) | (Hessel et al., 2017, Tab 6) |
| distrib-dqn | | (Hessel et al., 2017, Tab 6) |
| prior-ddqn | | (Wang et al., 2016, Tab 2) |
| prior-dqn | | (Wang et al., 2016, Tab 2) |
| prior-duel | | (Wang et al., 2016, Tab 2) |
| popart | (Hessel et al., 2018) | (Hessel et al., 2018, Tab 1) |
| dueling-ddqn | (Wang et al., 2016) | (Wang et al., 2016, Tab 2) |
| ddqn | (van Hasselt et al., 2015) | (Wang et al., 2016, Tab 2) |
| noisy-dqn | | (Hessel et al., 2017, Tab 6) |
| human | | (Hessel et al., 2017, Tab 6) |
| dqn | (Mnih et al., 2015) | (Hessel et al., 2017, Tab 6) |
| random | | (Hessel et al., 2017, Tab 6) |

(a) Atari agents and data reference                    (b) Agent Ratings

Figure 4: RL agents on Atari learning environments. The agents are rated in two gamification regimes: two-player (2P) zero-sum agent vs task, and three-player (3P) agent vs agent vs task. We evaluate using uniform and deviation ratings. The agents are ordered according to their uniform rating. We normalized all the ratings to be between $-1$ and $0$ (higher is better).

general-sum interactions between the task player and the agents. It is intended to only rating agents on hard but solvable tasks.

The normalized 2P uniform and 3P uniform ratings are identical, because after normalization the transform from the 2P to 3P game is linear. The uniform ratings are roughly ordered in terms of publication date, suggesting that decisions to publish are influenced by whether models outperform the current state of the art according to a uniform rating. Note that human performance is evaluated third last after `random` and `dqn` with the uniform rating.

The deviation ratings paint a more sophisticated picture. 2P deviation ranks four top agents equally, while the 3P deviation rating ranks the top three agents equally. By studying Table 3, we can see why this may be the case. `r2d2(bandit)` does well on `solaris`, `agent57` does well on `pitfall`, and `muzero` does well on `asteroids` and `beam-rider`. In particular these agents do much better on these tasks than the other top agents, awarding them joint first place according to deviation ratings. Deviation ratings also seem to reduce all the older agents to very small ratings because the evaluation is performed on difficult tasks that the earlier agents could not solve, therefore the deviation rating scheme adapts over to rate agents competently on hard tasks that are still solvable by at least some agents.

Additionally, there are a number of outliers. The ranking of `human` increases from 18th under uniform to 7th under 3P deviation. This is interesting because `human` has a distinct architecture compared to the other agents, and although is outclassed according the the uniform ratings (where they likely get lost amongst tasks that favour twitchy reflexes), `human` still does relatively well on tasks that the RL agents struggle with. The other outliers, `muzero2` and `ngu`, used search and intrinsic rewards respectively, which probably enabled them to fill niches that the other agents where not at the time.

| | r2d2 (bandit) | agent57 | muzero | r2d2 | r2d2 (retrace) | ngu | muesli | muzero2 | rainbow | distrib-dqn | prior-ddqn | prior-dqn | prior-duel | popart | dueling-ddqn | ddqn | noisy-dqn | human | dqn | random |
|---|---|---|---|---|---|---|---|---|---|---|---|---|---|---|---|---|---|---|---|---|
| asteroids | 0.063 | 0.022 | 1.000 | 0.058 | 0.051 | 0.037 | 0.071 | 0.075 | 0.000 | 0.000 | 0.000 | 0.000 | 0.000 | 0.000 | 0.000 | 0.000 | 0.001 | 0.007 | 0.000 | 0.000 |
| beam-rider | 0.086 | 0.066 | 1.000 | 0.054 | 0.027 | 0.017 | 0.063 | 0.070 | 0.004 | 0.003 | 0.005 | 0.005 | 0.007 | 0.002 | 0.003 | 0.003 | 0.003 | 0.004 | 0.002 | 0.000 |
| pitfall | 0.019 | 1.000 | 0.019 | 0.019 | 0.019 | 0.821 | 0.019 | 0.019 | 0.019 | 0.019 | 0.019 | 0.000 | 0.019 | 0.019 | 0.019 | 0.017 | 0.018 | 0.357 | 0.004 | 0.007 |
| solaris | 1.000 | 0.656 | 0.000 | 0.166 | 0.097 | 0.107 | 0.044 | 0.065 | 0.052 | 0.083 | 0.025 | 0.063 | 0.001 | 0.067 | 0.033 | 0.045 | 0.047 | 0.182 | 0.051 | 0.018 |
| ms-pacman | 0.256 | 0.262 | 1.000 | 0.206 | 0.184 | 0.199 | 0.267 | 0.325 | 0.021 | 0.014 | 0.018 | 0.026 | 0.012 | 0.019 | 0.025 | 0.010 | 0.009 | 0.027 | 0.011 | 0.000 |
| tutankham | 0.202 | 1.000 | 0.205 | 0.172 | 0.194 | 0.080 | 0.103 | 0.131 | 0.098 | 0.102 | 0.032 | 0.082 | 0.100 | 0.074 | 0.085 | 0.088 | 0.094 | 0.067 | 0.024 | 0.000 |
| zaxxon | 0.511 | 0.344 | 1.000 | 0.504 | 0.158 | 0.178 | 0.090 | 0.147 | 0.031 | 0.025 | 0.019 | 0.014 | 0.019 | 0.020 | 0.018 | 0.014 | 0.013 | 0.013 | 0.007 | 0.000 |
| alien | 0.626 | 0.401 | 1.000 | 0.539 | 0.308 | 0.420 | 0.188 | 0.182 | 0.012 | 0.005 | 0.009 | 0.005 | 0.005 | 0.004 | 0.006 | 0.005 | 0.003 | 0.009 | 0.002 | 0.000 |
| private-eye | 0.406 | 0.795 | 0.152 | 0.187 | 0.345 | 1.000 | 0.103 | 0.076 | 0.042 | 0.151 | 0.002 | 0.002 | 0.002 | 0.003 | 0.001 | 0.001 | 0.039 | 0.693 | 0.001 | 0.000 |
| bank-heist | 0.704 | 0.599 | 0.033 | 1.000 | 0.434 | 0.519 | 0.031 | 0.368 | 0.035 | 0.027 | 0.029 | 0.027 | 0.039 | 0.028 | 0.041 | 0.026 | 0.034 | 0.019 | 0.011 | 0.000 |
| qbert | 1.000 | 0.747 | 0.093 | 0.992 | 0.559 | 0.616 | 0.202 | 0.110 | 0.043 | 0.022 | 0.024 | 0.021 | 0.024 | 0.007 | 0.025 | 0.019 | 0.019 | 0.017 | 0.017 | 0.000 |
| assault | 0.764 | 0.466 | 1.000 | 0.868 | 0.318 | 0.296 | 0.256 | 0.205 | 0.097 | 0.040 | 0.054 | 0.052 | 0.078 | 0.061 | 0.031 | 0.036 | 0.035 | 0.004 | 0.028 | 0.000 |
| frostbite | 0.489 | 0.857 | 1.000 | 0.707 | 0.019 | 0.450 | 0.478 | 0.650 | 0.015 | 0.006 | 0.005 | 0.007 | 0.012 | 0.005 | 0.007 | 0.003 | 0.001 | 0.007 | 0.001 | 0.000 |
| krull | 0.842 | 0.865 | 0.925 | 1.000 | 0.510 | 0.516 | 0.113 | 0.168 | 0.025 | 0.029 | 0.030 | 0.028 | 0.030 | 0.028 | 0.028 | 0.022 | 0.026 | 0.004 | 0.024 | 0.000 |
| star-gunner | 1.000 | 0.841 | 0.550 | 0.925 | 0.421 | 0.450 | 0.214 | 0.158 | 0.127 | 0.069 | 0.056 | 0.063 | 0.125 | 0.000 | 0.089 | 0.060 | 0.034 | 0.010 | 0.054 | 0.000 |
| name-this-game | 0.876 | 0.336 | 1.000 | 0.466 | 0.464 | 0.151 | 0.663 | 0.683 | 0.070 | 0.069 | 0.072 | 0.064 | 0.086 | 0.088 | 0.062 | 0.054 | 0.039 | 0.037 | 0.038 | 0.000 |
| centipede | 0.783 | 0.355 | 1.000 | 0.598 | 0.636 | 0.514 | 0.750 | 0.744 | 0.005 | 0.006 | 0.003 | 0.002 | 0.005 | 0.041 | 0.005 | 0.003 | 0.002 | 0.009 | 0.002 | 0.000 |
| berzerk | 0.904 | 0.715 | 1.000 | 0.754 | 0.855 | 0.530 | 0.517 | 0.226 | 0.028 | 0.015 | 0.017 | 0.014 | 0.038 | 0.013 | 0.016 | 0.013 | 0.008 | 0.029 | 0.005 | 0.000 |
| gravitar | 1.000 | 0.911 | 0.312 | 0.822 | 0.671 | 0.699 | 0.550 | 0.515 | 0.060 | 0.024 | 0.008 | 0.018 | 0.003 | 0.015 | 0.020 | 0.011 | 0.013 | 0.152 | 0.014 | 0.000 |
| road-runner | 0.967 | 0.396 | 1.000 | 1.000 | 0.189 | 0.247 | 0.363 | 0.904 | 0.101 | 0.103 | 0.102 | 0.094 | 0.101 | 0.078 | 0.113 | 0.072 | 0.068 | 0.013 | 0.064 | 0.000 |
| hero | 0.425 | 1.000 | 0.424 | 0.341 | 0.474 | 0.621 | 0.318 | 0.319 | 0.482 | 0.289 | 0.230 | 0.194 | 0.176 | 0.116 | 0.174 | 0.168 | 0.035 | 0.262 | 0.171 | 0.000 |
| wizard-of-wor | 0.929 | 0.798 | 1.000 | 0.910 | 0.679 | 0.617 | 0.472 | 0.524 | 0.088 | 0.079 | 0.050 | 0.022 | 0.060 | 0.000 | 0.037 | 0.036 | 0.025 | 0.022 | 0.011 | 0.000 |
| crazy-climber | 1.000 | 0.772 | 0.623 | 0.749 | 0.434 | 0.474 | 0.229 | 0.230 | 0.220 | 0.233 | 0.240 | 0.181 | 0.211 | 0.152 | 0.185 | 0.148 | 0.150 | 0.035 | 0.139 | 0.000 |
| battle-zone | 1.000 | 0.941 | 0.855 | 0.963 | 0.852 | 0.820 | 0.416 | 0.321 | 0.060 | 0.039 | 0.036 | 0.029 | 0.033 | 0.006 | 0.035 | 0.030 | 0.030 | 0.035 | 0.028 | 0.000 |
| yars-revenge | 1.000 | 0.999 | 0.552 | 1.000 | 0.999 | 0.998 | 0.557 | 0.185 | 0.100 | 0.014 | 0.013 | 0.008 | 0.067 | 0.018 | 0.047 | 0.009 | 0.006 | 0.052 | 0.015 | 0.000 |
| chopper-command | 1.000 | 1.000 | 0.991 | 1.000 | 1.000 | 1.000 | 0.101 | 0.494 | 0.016 | 0.012 | 0.004 | 0.008 | 0.012 | 0.000 | 0.010 | 0.005 | 0.009 | 0.007 | 0.005 | 0.000 |
| ice-hockey | 0.997 | 0.763 | 0.798 | 1.000 | 0.997 | 0.082 | 0.369 | 0.522 | 0.125 | 0.127 | 0.117 | 0.127 | 0.110 | 0.072 | 0.119 | 0.087 | 0.093 | 0.123 | 0.095 | 0.000 |
| space-invaders | 0.913 | 0.654 | 1.000 | 0.904 | 0.484 | 0.646 | 0.801 | 0.419 | 0.251 | 0.091 | 0.102 | 0.037 | 0.204 | 0.033 | 0.085 | 0.032 | 0.027 | 0.020 | 0.021 | 0.000 |
| amidar | 1.000 | 0.947 | 0.914 | 0.968 | 0.918 | 0.586 | 0.691 | 0.034 | 0.164 | 0.040 | 0.065 | 0.059 | 0.073 | 0.025 | 0.075 | 0.057 | 0.051 | 0.055 | 0.031 | 0.000 |
| defender | 0.870 | 0.806 | 1.000 | 0.824 | 0.811 | 0.814 | 0.749 | 0.647 | 0.062 | 0.042 | 0.025 | 0.034 | 0.046 | 0.010 | 0.047 | 0.039 | 0.024 | 0.019 | 0.025 | 0.000 |
| venture | 0.861 | 1.000 | 0.000 | 0.780 | 0.767 | 0.666 | 0.802 | 0.330 | 0.002 | 0.422 | 0.329 | 0.021 | 0.018 | 0.447 | 0.189 | 0.037 | 0.000 | 0.453 | 0.062 | 0.000 |
| time-pilot | 0.966 | 0.849 | 1.000 | 0.952 | 0.950 | 0.771 | 0.751 | 0.867 | 0.020 | 0.009 | 0.017 | 0.012 | 0.008 | 0.003 | 0.017 | 0.010 | 0.005 | 0.004 | 0.003 | 0.000 |
| kangaroo | 0.488 | 0.642 | 0.448 | 0.387 | 0.405 | 1.000 | 0.376 | 0.372 | 0.391 | 0.344 | 0.387 | 0.432 | 0.047 | 0.351 | 0.396 | 0.347 | 0.323 | 0.080 | 0.193 | 0.000 |
| seaquest | 1.000 | 1.000 | 1.000 | 1.000 | 1.000 | 1.000 | 0.816 | 0.501 | 0.016 | 0.005 | 0.044 | 0.026 | 0.001 | 0.011 | 0.050 | 0.016 | 0.002 | 0.042 | 0.006 | 0.000 |
| phoenix | 1.000 | 0.917 | 0.964 | 0.884 | 0.947 | 0.976 | 0.813 | 0.755 | 0.109 | 0.034 | 0.032 | 0.018 | 0.070 | 0.005 | 0.023 | 0.012 | 0.016 | 0.007 | 0.008 | 0.000 |
| kung-fu-master | 1.000 | 0.772 | 0.765 | 0.944 | 0.852 | 0.806 | 0.503 | 0.554 | 0.194 | 0.160 | 0.162 | 0.147 | 0.180 | 0.128 | 0.127 | 0.110 | 0.127 | 0.084 | 0.096 | 0.000 |
| asterix | 1.000 | 0.992 | 0.999 | 1.000 | 0.999 | 0.997 | 0.316 | 0.919 | 0.428 | 0.401 | 0.041 | 0.031 | 0.375 | 0.019 | 0.028 | 0.017 | 0.012 | 0.008 | 0.004 | 0.000 |
| bowling | 0.585 | 0.962 | 1.000 | 0.870 | 0.991 | 0.811 | 0.708 | 0.561 | 0.029 | 0.215 | 0.167 | 0.105 | 0.100 | 0.333 | 0.179 | 0.190 | 0.229 | 0.581 | 0.115 | 0.000 |
| atlantis | 0.992 | 0.912 | 1.000 | 0.982 | 0.991 | 0.991 | 0.813 | 0.676 | 0.490 | 0.157 | 0.250 | 0.207 | 0.230 | 0.197 | 0.222 | 0.056 | 0.190 | 0.010 | 0.161 | 0.000 |
| robotank | 1.000 | 0.882 | 0.909 | 0.906 | 0.997 | 0.066 | 0.401 | 0.584 | 0.417 | 0.367 | 0.398 | 0.426 | 0.178 | 0.438 | 0.445 | 0.444 | 0.362 | 0.068 | 0.435 | 0.000 |
| gopher | 0.995 | 0.903 | 1.000 | 0.968 | 0.919 | 0.914 | 0.801 | 0.931 | 0.539 | 0.220 | 0.375 | 0.248 | 0.800 | 0.430 | 0.119 | 0.112 | 0.114 | 0.017 | 0.065 | 0.000 |
| double-dunk | 1.000 | 0.998 | 0.999 | 0.999 | 1.000 | 0.140 | 0.366 | 1.000 | 0.430 | 0.347 | 0.549 | 0.871 | 0.143 | 0.167 | 0.439 | 0.308 | 0.394 | 0.052 | 0.282 | 0.000 |
| video-pinball | 1.000 | 0.993 | 0.982 | 1.000 | 0.965 | 0.974 | 0.686 | 0.922 | 0.534 | 0.479 | 0.407 | 0.282 | 0.479 | 0.056 | 0.098 | 0.310 | 0.271 | 0.018 | 0.197 | 0.000 |
| skiing | 1.000 | 0.987 | 0.001 | 0.467 | 0.590 | 0.219 | 0.443 | 0.000 | 0.652 | 0.575 | 0.769 | 0.765 | 0.384 | 0.628 | 0.809 | 0.802 | 0.524 | 0.981 | 0.648 | 0.493 |
| tennis | 1.000 | 0.997 | 0.498 | 0.664 | 1.000 | 0.729 | 0.749 | 0.498 | 0.498 | 0.992 | 0.498 | 0.498 | 0.498 | 0.751 | 0.605 | 0.021 | 0.498 | 0.324 | 0.753 | 0.000 |
| breakout | 1.000 | 0.915 | 1.000 | 0.999 | 0.995 | 0.724 | 0.915 | 0.900 | 0.482 | 0.708 | 0.440 | 0.432 | 0.422 | 0.397 | 0.398 | 0.483 | 0.530 | 0.033 | 0.445 | 0.000 |
| demon-attack | 1.000 | 0.994 | 1.000 | 0.999 | 1.000 | 0.998 | 0.900 | 0.999 | 0.772 | 0.487 | 0.499 | 0.506 | 0.441 | 0.422 | 0.403 | 0.172 | 0.013 | 0.083 | | 0.000 |
| surround | 1.000 | 0.975 | 1.000 | 1.000 | 0.998 | 0.034 | 0.950 | 1.000 | 0.985 | 0.810 | 0.605 | 0.945 | 0.560 | 0.375 | 0.720 | 0.355 | 0.335 | 0.825 | 0.220 | 0.000 |
| fishing-derby | 0.996 | 0.977 | 1.000 | 0.982 | 0.982 | 0.691 | 0.780 | 0.879 | 0.673 | 0.551 | 0.667 | 0.717 | 0.727 | 0.748 | 0.755 | 0.586 | 0.544 | 0.290 | 0.475 | 0.000 |
| enduro | 0.998 | 0.994 | 1.000 | 0.999 | 0.996 | 0.880 | 0.991 | 0.992 | 0.892 | 0.948 | 0.905 | 0.879 | 0.968 | 0.840 | 0.948 | 0.509 | 0.474 | 0.361 | 0.306 | 0.000 |
| freeway | 1.000 | 0.959 | 0.971 | 0.968 | 0.985 | 0.844 | 0.971 | 1.000 | 1.000 | 0.988 | 0.968 | 0.991 | 0.971 | 0.982 | 0.000 | 0.979 | 0.941 | 0.871 | 0.906 | 0.000 |
| boxing | 1.000 | 1.000 | 1.000 | 0.993 | 1.000 | 0.997 | 0.990 | 1.000 | 0.996 | 0.981 | 0.988 | 0.956 | 0.989 | 0.993 | 0.994 | 0.916 | 0.833 | 0.120 | 0.880 | 0.000 |
| pong | 1.000 | 0.992 | 1.000 | 1.000 | 0.999 | 0.972 | 0.976 | 1.000 | 0.998 | 0.995 | 0.993 | 0.990 | 0.998 | 0.990 | 1.000 | 0.998 | 1.000 | 0.847 | 0.964 | 0.000 |

Table 3: Normalized RL agent rating amalgamation from sources described in Table 4a. The rows and columns are ordered according to uniform rating on the agent vs task regime. The data are normalized between zero and one for each game.

