# OpenReview forum: "Deviation Ratings: A general, clone invariant rating method"
_ICLR.cc/2025/Conference — Submitted to ICLR 2025_

### Official Review · Reviewer_DCJN · 2024-10-23

**Soundness:** 4
**Presentation:** 2
**Contribution:** 1
**Rating:** 6
**Confidence:** 2

**Summary:**

This paper focuses on the problem of rating different strategies in a normal-form game. According to the authors, many real-world multi-agent or multi-task evaluation scenarios can be modeled as normal-form games, including, for example, the evaluation of large language models, which motivates the study. Prior work on this topic has either focused on two-player zero-sum games or has not been clone-invariant. The authors claim that clone invariance, i.e., copying strategies does not change the ratings, is an important desideratum for the rating scheme. The authors then propose a new rating scheme that is clone-invariant and satisfies a set of other desirable properties. The proposed rating scheme is evaluated in several environments, including rating strategies in simple tabular games, evaluating large language models, and evaluating strategies in well-known benchmark games. The authors claim that the evaluation results show the merits of the proposed rating scheme.

**Strengths:**

1. The preliminary and related work sections are well-written and provide a good overview of the existing work.
2. In the theoretical sections, the results are clearly stated and sound.
3. The environments used for evaluation are diverse and cover a wide range of scenarios.

**Weaknesses:**

I have two major concerns about this paper that prevent me from recommending it for acceptance:

1. **Evaluation.** Other than the stated desiderata, it is not clear to me why the proposed rating scheme is better than existing ones. The results in the evaluation section (Figures 2 and 3) do not seem to demonstrate that the proposed rating scheme is better than existing ones. For example, in Figure 2(a), the proposed rating scheme and the benchmarks perform similarly to me, and there are no quantitative measures of their performance to compare them. From my perspective, the motivation for clone invariance is not so strong that it can be used as the main criterion for evaluating the rating scheme.
2. **Interpretation of the results.** In the introduction, the authors frame the results in a way that suggests that the results are relevant to many real-world multi-agent or multi-task evaluation scenarios. However, since the evaluation fails to convince me that the proposed rating scheme is even better than existing ones, the relevance of the results to real-world scenarios is further weakened. For example, even if one can interpret the results in Figure 3(a) as showing that the proposed rating scheme is better than the benchmarks in reducing the equilibrium gap, there are much better ways to do this, such as using no-regret learning algorithms.

**Post rebuttal:** The authors clarified the motivation for clone invariance in the rebuttal in a compelling way. Moreover, the authors emphasized a potential use case, and accounted for the lack of quantitative evaluation. I have increased my rating, but lowered my confidence for this paper. Given the current form, I believe this paper needs significant reframing for readers to understand its motivation and value. I will be okay with this paper being accepted if the authors would revise the paper according to the detailed discussion we had.

**Questions:**

1. How can the authors demonstrate that the proposed rating scheme is better than existing ones?
2. Why is the proposed rating scheme relevant to real-world scenarios? Can you provide more concrete examples?

### Minor comments

1. Line 56: "Yoa’s Principle" -> "Yao’s Principle".
2. Line 111: "Rankings can be inferred from ratings, and are therefore more general" - Did you mean the other way around?
3. Line 132: "$\tilde{G}(a_p, a_{-p})$" - Should be "$\tilde{G_p}(a_p, a_{-p})$"?

---

> ### Author Response · Authors · 2024-11-19
> **DCJN Rebuttal 1**
>
> Thank you for checking the proofs and your vote of confidence in their correctness. Furthermore, thank you for the line-by-line issues, we have fixed them. Your comments have helped improve the clarity and motivation of our paper significantly and we hope our replies help with your impression of the paper too.
>
> > From my perspective, the motivation for clone invariance is not so strong that it can be used as the main criterion for evaluating the rating scheme.
>
> We took the importance of clone-invariance for granted when writing the paper. Please let us correct that with four key arguments: a) maximum inclusivity, b) resistance to attacks, c) hill-climbability d) wealth of literature / reinvention. We will include these additional arguments in the paper.
>
> [Maximum Inclusivity] Balduzzi [1] states the maximum inclusivity elegantly: “Evaluation is often based on metrics like average performance or Elo ratings. Unfortunately, two (or two hundred) tasks or agents that look different may test/exhibit identical skills. Overrepresenting particular tasks or agents introduces biases into averages and Elo – biases that can only be detected post hoc. Humans must therefore decide which tasks or agents to retain, to prevent redundant agents or tasks from skewing results. At present, evaluation is not automatic and does not scale. To be scalable and automatic, an evaluation method should always benefit from including additional agents and tasks. Moreover, it should adjust automatically and gracefully to redundant data.” Therefore clone-invariance allows us to be maximally inclusive with training data which enables scalable evaluation.
>
> [Resistance to Attacks] Redundancy to clones means that the rating is less susceptible to a “clone attack”, where the frequency of specific tasks are inflated so that average ratings of models can be manipulated. This is particularly useful in scenarios where the tasks cannot be easily curated, and can be submitted by anyone, such as in Chatbot Arena.
>
> [Hill-Climbable] Ratings are often optimised against (albeit in a coarse, perhaps human-driven, outer-loop). Because uniform and Elo are so susceptible to the distribution of the data this also makes them tricky to optimise against. For example, models may favour getting ever-stronger on well-represented tasks, or against other common models, and may ignore niche tasks. Optimising against deviation-ratings, will result in more holistic improvement (which we argue in Figure 3).
>
> [Popularity] In the two-player zero-sum setting, clone-invariant rating schemes have been continuously explored and re-invented, in multiple disciplines (economics, mathematics, social choice, philosophy, computer science, political science, game theory) over the years [1,2,3,4,5,6,7,8,9,10]. In particular, in social choice theory, “clone invariance” is often called the “independence of clones criterion” [11], and is researched in the context of voting schemes where cloned candidates can spoil an election. This highlights a huge appetite for this property, and is evidence that it is perhaps a fundamental property.
>
> Finally, we would like to point out that deviation ratings preserve all the other important desiderata. Clone-invariance is not coming at the expense of any other important property (dominance preserving, offset invariant).
>
> [1] David Balduzi (2018), “Re-evaluating evaluation”.
>
> [2] G. Kreweras. (1960), “Aggregation of preference orderings”.
>
> [3] Vincent Conitzer (2024) “Social Choice Should Guide AI Alignment in Dealing with Diverse Human Feedback”
>
> [4] F. Brandl, F. Brandt, and H. G. Seedig. (2016) Consistent probabilistic social choice.
>
> [5] Fishburn. (1984) “Probabilistic social choice based on simple voting comparisons”.
>
> [6] Yao, Andrew (1977), "Probabilistic computations: Toward a unified measure of complexity".
>
> [7] Laffond, J.-F. Laslier, and M. Le Breton. (1993) “The bipartisan set of a tournament game”.
>
> [8] D. C. Fisher and J. Ryan. (1995) “Tournament games and positive tournaments”.
>
> [9] D. S. Felsenthal and M. Machover. (1992) “After two centuries should Condorcet’s voting procedure be implemented?”
>
> [10] R. L. Rivest and E. Shen. (2010) “An optimal single-winner preferential voting system based on game theory.”
>
> [11] T. N. Tideman (1987). “Independence of clones as a criterion for voting rules”.

---

> ### Author Response · Authors · 2024-11-19
> **DCJN Rebuttal 2**
>
> > Why is the proposed rating scheme relevant to real-world scenarios?
> > Can you provide more concrete examples?
>
> Current LLM evaluation could benefit from “maximum inclusivity/scalability”, “clone-attack-proofness”,  and “hill-climbability”. LLMs are evaluated on public leaderboards, like Chatbot Arena, where anyone can submit a prompt, and models are evaluated with Elo which closely approximates a uniform average. The prompts cover a wide array of tasks/skills.
>
> [maximum inclusivity/scalability] If the rating method was clone-invariant then we can include all evaluation data without having to worry about curation or any biases in the distribution, allowing us to simply scale the evaluation data. Deviation ratings are maximally inclusive, Elo/uniform is not.
>
> [clone-attack-proofnes] Let’s say that someone submits a model to the leaderboard, and knows that their model is state-of-the-art at one particular task (say SQL query problems), but poor at everything else. They could arrange that many SQL prompts be submitted to the leaderboard, artificially inflating their own model’s score. Even if you assumed no attacks were taking place, there is nothing fundamental about the distribution prompts that are being submitted on ChatbotArena. Deviation ratings are immune to clone attack.
>
> [hill-climbing] People and companies are continuously hill-climbing on the ratings produced by leaderboards. With poor ratings, users may choose weaker models for their use cases and funding/resources may be assigned sub-optimally. Ratings are now very high-stake signals. Deviation ratings are better to hill-climb on than the alternatives.
>
> > The results in the evaluation section (Figures 2 and 3) do not seem to demonstrate that the proposed rating scheme is better than existing ones. For example, in Figure 2(a), the proposed rating scheme and the benchmarks perform similarly to me
>
> There are only so many reasonable ways evaluation data can be aggregated into a scalar so the fact, coarsely, that there are no extreme differences in the ratings is not surprising. The differences will be subtle, yet important. Figure 4 in the appendix shows an example with more extreme rating differences that you may be interested in.
>
> On the differences in Figure 2, focus on the order of the ratings. Uniform and Elo have the same order as each other and are not meaningfully different. However the deviation ratings have a meaningfully different ordering. In particular, the top four models have equal rating, an interesting property that emerges in clone-invariant game-theoretic rating - a significant difference to Elo or uniform. Furthermore, there is significant movement in the top 10 with models shifting up to 5 positions in the rankings compared to uniform/Elo.
>
> We therefore strongly disagree that deviation ratings result in similar ratings to Elo and uniform.

---

> ### Author Response · Authors · 2024-11-19
> **DCJN Rebuttal 3**
>
> > and there are no quantitative measures of their performance to compare them.
> > How can the authors demonstrate that the proposed rating scheme is better than existing ones?
>
> We are not familiar with any metrics to compare rating methods. If you are familiar with one, we would happily assess them under this metric. We sympathise with the desire for quantitative measures of performance. However, choosing a metric for ratings is often fraught or circular. For example, we could invent a measure of clone-invariance, but then we would have to defend the definition of the metric, and it would appear contrived. And if we came up with a metric other than the property we are measuring, why not just optimise for that metric directly to define another rating scheme?
>
> As a result, the rating and game theory literature tends to favour axiomatic/qualitative arguments (*). The approach is: shortlist a bunch of desirable properties, try to invent a rating that has these properties, prove the properties hold, and then show the ratings with real data to build intuition. We have pursued a similar approach in this paper. We have many qualitative measures, and have gone to great lengths to define and prove them (summarised in Table 1). Concretely: dominant invariant, clone-invariant, mixture invariant, offset invariant, n-player, and general-sum. To our knowledge this is the only known rating method with these properties.
>
> We acknowledge that a lack-of-quantitive-evaluation explanation was not made properly in the paper and may not be obvious to readers unfamiliar with the rating literature. We have amended the paper to include these arguments and renamed the “Evaluation Studies” to “Illustrative Studies” to better reflect the intention of the section.
>
> (*) For example, Nash bargaining [1] has the properties of Pareto efficiency, symmetry, invariance to affine transformations, and independence of irrelevant alternatives. Arrow's Impossibility Theorem in Social Choice [2] has unrestricted domain, non-dictatorship, Pareto efficiency, and independence of irrelevant alternatives. Shapley Value [3] has efficiency, zero dummy player payoff, symmetry, and additivity. The core [4] has efficiency, and coalitional rationality.
>
> [1] Nash, John F. (1950). "The Bargaining Problem". Econometrica. 18 (2): 155–162.
>
> [2] Arrow, Kenneth J. (1951). Social Choice and Individual Values. New York: Wiley.
>
> [3] Shapley, Lloyd S. (1953). "A value for n-person games". In Kuhn, H.; Tucker, A. W. (eds.). Contributions to the Theory of Games II. Princeton: Princeton University Press. pp. 307–317.
>
> [4] Gillies, D. B. (1959). "Solutions to general non-zero-sum games". In Tucker, A. W.; Luce, R. D. (eds.). Contributions to the Theory of Games IV. Annals of Mathematics Studies. Vol. 40. Princeton: Princeton University Press. pp. 47–85.
>
> > For example, even if one can interpret the results in Figure 3(a) as showing that the proposed rating scheme is better than the benchmarks in reducing the equilibrium gap, there are much better ways to do this, such as using no-regret learning algorithms.
>
> The purpose of this section was not to propose a learning algorithm. Instead it was to simulate a much larger model improvement process, for example where iterations are very expensive, decisions are made by people, and informed by evaluations. In such a setting we cannot deploy a learning algorithm but can provide better evaluations. We approximate this process by assuming humans pick the highest evaluated models, the evaluations are ratings, and the structured problem space the models are solving is some OpenSpiel game (some system where there are strategic trade-offs to be made). We are attempting to show that with improved ratings this process can be improved.
>
> > Line 111: "Rankings can be inferred from ratings, and are therefore more general" - Did you mean the other way around?
>
> Ratings are scalars and rankings are the natural numbers (corresponding to an ordering).

---

> > ### Comment · Reviewer_DCJN · 2024-11-19
> >
> > Thank you for your detailed reply to my review.
> >
> > I have read through your response and have a few questions for clarification.
> >
> > **1. Motivation for clone invariance.**
> > After reading your reply, the motivation for clone invariance now feels more compelling to me. Am I correct in understanding that the primary motivation for the entire study is the significance of clone invariance?
> >
> > **2. Evaluation.**
> > Your response suggests that the proposed method outperforms existing approaches primarily because it satisfies a set of desiderata—such as clone invariance and general-sumness—that are not met by other methods. Consequently, while there are no quantitative measures of performance, this conceptual improvement alone provides a strong basis for evaluating the proposed method. Is this interpretation correct?
> >
> > **3. Rankings and ratings.**
> > Regarding Line 111: "Rankings can be inferred from ratings, and are therefore more general" seems to imply that "Rankings are more general." However, I believe you intended to convey that "Ratings are more general." If so, this might need clarification, although I consider it a minor point.

---

> > > ### Author Response · Authors · 2024-11-20
> > >
> > > 1.
> > >
> > > The primary motivation of the paper is proposing a principled n-player, general-sum rating that respects several desiderata, of which clone invariance is the most difficult to achieve, and therefore is given prominent focus in the paper. However, other properties are important too. For example, a rating system that was not dominance preserving could generate very counterintuitive ratings. We are proposing a new rating algorithm that we think is useful and, where possible, should be used by practitioners instead of Elo or uniform average.
> > >
> > > Section 5 serves multiple purposes. It is about motivating clone-invariance, but also demonstrating how the rating can be used by practitioners, including how to construct games from data and additional visualizations that can be produced.
> > >
> > > Figure 1 is primarily about motivating clone-invariance.
> > >
> > > Figure 2 also motivates clone-invariance (top-four having equal rating property). We think this will be particularly interesting to people who are familiar with LLM models. People familiar with LLM models will understand that the large Claude, Gemini, LLama, and GPT are all very strong models and each have domains they are best in. Figure 2 also shows other properties, most notably that our proposed deviation gain rating system also is dominance preserving. The section also describes a way of constructing a 3-player game from model vs task data, and shows that not only are models rated sensibly, but the rating focuses primarily on 4 tasks which separate the models. As evaluation tasks are difficult to maintain we think that practitioners would also be excited by the tasks also being rated. We didn’t include the task ratings in Figure 2, but could add them if you think it will be interesting?
> > >
> > > Figure 3 is primarily about motivating the “hill-climbing” argument.
> > >
> > >
> > > 2.
> > >
> > > Correct - we are primarily motivating the proposed method in an axiomatic/qualitative way. There is no “ground-truth” rating method. An axiomatic approach gives the community the greatest clarity as to what the strengths of a rating method are as well as how and where it can be used. We think the combination of clone-invariance, dominance preserving, n-player, and general-sum is particularly compelling. We also hope Section 5 supports our case.
> > >
> > >
> > > 3.
> > >
> > > Understood, you are right. We will change to simply “Rankings can be inferred from ratings”.

---

> > > > ### Comment · Reviewer_DCJN · 2024-11-20
> > > >
> > > > I think you clarified the motivation for clone invariance in the rebuttal in a compelling way. Moreover, you emphasized a potential use case, and accounted for the lack of quantitative evaluation. I have increased my rating, but lowered my confidence for this paper. Given the current form, I believe this paper needs significant reframing for readers to understand its motivation and value. I will be okay with this paper being accepted if you would revise the paper according to the detailed discussion we had. Specifically, I would expect the motivation for clone invariance and the specific use on LLM evaluations to be emphasized in the introduction as the main contribution, and reframe the evaluation studies to account for the situation of the lack of quantitative metrics. I believe this would make the value of this paper a lot clearer.

---

> > > > > ### Author Response · Authors · 2024-11-25
> > > > >
> > > > > Thank for you the valuable feedback on reframing. We have rewrote the introduction and reframed the evaluation studies and believe the paper to be stronger for it. The latest draft has been updated on openreview.

---

### Official Review · Reviewer_v7Bf · 2024-11-04

**Soundness:** 3
**Presentation:** 2
**Contribution:** 3
**Rating:** 6
**Confidence:** 2

**Summary:**

The paper introduces a new game-theoretic rating method for strategies in N-player, general-sum settings, called "deviation rating", which is clone-invariant. The authors provide an algorithm for computing the rating efficiently with linear programming and conduct experiments to evaluate the proposed rating method on three applications, namely Shapley’s game, LLM rating, and model improvement.

**Strengths:**

+ The paper proposes a clone-invariant rating method that is applicable to N-player general-sum strategic interactions, with proofs on existence, uniqueness, and other properties.
+ Extensive experiments over a wide range of applications.

**Weaknesses:**

- Insufficient implementation details: computational resources, LP solver, actual number of iterations needed (with respect to Algorithm 1), runtimes, etc.
- Presentations: There should be some outline for the paper and transitions between sections. Also, in the abstract, an overview on deviation rating as well as its evaluation (as in Section 5) is missing.

**Questions:**

Please include the missing implementation details and polish the paper presentations, and if possible, please add a dedicated Related Work section (where existing rating methods as listed in Section 3.2 and other relevant lines of work are more thoroughly discussed).

---

> ### Author Response · Authors · 2024-11-19
> **v7Bf Rebuttal**
>
> Thank you for your review and your suggestions. We have incorporated some of your suggestions and are actively working on others.
>
> > Insufficient implementation details: computational resources, LP solver, actual number of iterations needed (with respect to Algorithm 1), runtimes, etc.
>
> We apologize for these omissions. We have included details of the solvers, and their parameters in the latest draft version. Runtimes will add runtimes in a future revision.
>
> >Presentations: There should be some outline for the paper and transitions between sections. Also, in the abstract, an overview on deviation rating as well as its evaluation (as in Section 5) is missing.
>
> We have updated the abstract and added more signposting throughout the paper.
>
> > please add a dedicated Related Work section (where existing rating methods as listed in Section 3.2 and other relevant lines of work are more thoroughly discussed)
>
> We will add a larger related work section, including connections to social choice theory, in the appendix in a later revision.

---

> ### Author Response · Authors · 2024-11-27
>
> We have added more background on rating methods in Appendix A.

---

> > ### Comment · Reviewer_v7Bf · 2024-11-27
> >
> > I appreciate the efforts from the authors in incorporating my suggestions. However, given that the related work section contextualizing the contributions of this paper among existing literature is still missing, I will keep the score as it is.

---

> > > ### Author Response · Authors · 2024-11-27
> > >
> > > We have added such a section in Appendix Section A. Did you have a chance to look at that?

---

> > > > ### Comment · Reviewer_v7Bf · 2024-11-27
> > > >
> > > > How are the evolutionary-based ratings and other methods in A.4 and A.5 compared to the current work? Are they clone-invariant? If not, how do they address the issue of repeated strategies? Please compare and contrast them to the proposed method to further clarify the paper contribution.

---

> > > > > ### Author Response · Authors · 2024-12-02
> > > > >
> > > > > > How are the evolutionary-based ratings and other methods in A.4 and A.5 compared to the current work? Are they clone-invariant?
> > > > >
> > > > > The evolutionary-based method (alpha-Rank) is *not clone-invariant*. Their only relevance to the current work is that they are also game-theoretic inspired rating methods and they work in N-player general-sum games.
> > > > >
> > > > > TrueSkill and Glicko are *not clone-invariant* and are cited because they are other famous ratings. Otherwise their properties are not comparable to deviation ratings: they are not N-player or general-sum either.
> > > > >
> > > > > VasE encompasses a number of rating methods. Only one is clone-invariant (maximal lotteries) - which we have discussed in the social choice section.
> > > > >
> > > > > Unfortunately we cannot upload another draft at this stage -- but we have further refined this section with your helpful suggestions.

---

### Official Review · Reviewer_y7Ru · 2024-11-05

**Soundness:** 2
**Presentation:** 2
**Contribution:** 2
**Rating:** 5
**Confidence:** 3

**Summary:**

In this work, the authors address the problem of rating strategic interactions within a game-theoretic model. They propose a new rating scheme, called Deviation Ratings, which is clone-invariant (i.e., unaffected by repeated or similar actions) and is applicable to multi-player settings.

**Strengths:**

I do not see a clear strength in this work, apart from introducing a new game-theoretic rating system for multi-player settings with certain advantageous properties. However, this may not be a strong point, as similar concepts already exist in the literature. Furthermore, I am somewhat skeptical about the overall contribution, though I should note that I am not an expert in this field and would welcome the authors’ clarification regarding my concerns.

**Weaknesses:**

My primary concern is with the definition of the rating scheme itself. Strategies are evaluated based on the advantages of deviation strategies. However, since the rating is assessed under a correlated equilibrium, it would necessarily be negative (or $\leq \epsilon$) in the case of approximate CCE), which seems uninformative. For example, the rating scheme in [1] appears more intuitive. It is possible I have misunderstood some details; could the authors clarify?

Additionally, the extension of Nash averaging to general multi-player environments seems to have already been explored in [1]. In particular, [1] defines a rating scheme called Payoff Rating, where a strategy is evaluated based on its expected payoff under the correlated equilibrium $\sigma$, conditioned on that strategy. As I am not deeply familiar with this area, could the authors further elaborate on the connection between these two results, as the relationship is not clearly highlighted in this paper?

Regarding the rating definition, the authors state that they aim to minimize deviation gains, but it’s not intuitive why one would seek to **minimize** the so-called **gains** of a strategy.

**Questions:**

- **Line 88**: Since the concepts of WSCE and CE, as well as their approximate variants, are not directly relevant to the scope of this work, it might be better to exclude them, as their current presentation adds more confusion than clarity.
- **Line 56**: Typo in "Yoa’s principle."
- **Line 70**: Typo in "for the strictest."
- **Line 103**: In the equation for Nash equilibrium as a product distribution, $\sigma_p(a_1)$ should be $\sigma_1(a_1)$.
- **Line 152**: The rating for Nash averaging can be simplified as there are only two players, so the product notation can be removed.
- **Line 170**: "one alternative to the payoff rating…" The authors in [1] define a rating called Payoff Rating, which employs the same idea of payoff-based rating, potentially contradicting the authors’ claim here. This paragraph does not clearly differentiate between the “mass rating” type and the Payoff Rating, and the term "mass rating" itself lacks a clear definition.
- **Line 197**: How are “deviation gain statistics” defined?
- **Line 225**: Strategies are evaluated based on a correlated equilibrium computed from the deviation **gains** of players’ "uncovered" strategies. Intuitively, some properties may fail here, as they would be inherently dependent on the notion of correlated equilibrium.

**References**
[1] Game Theoretic Rating in N-player General-Sum Games with Equilibria

---

> ### Author Response · Authors · 2024-11-19
> **y7Ru Rebuttal**
>
> Thank you for your review and line-by-line points. We will work your suggestions into the paper and hope the explanations below will help your understanding.
>
> > as similar concepts already exist in the literature
> > Additionally, the extension of Nash averaging to general multi-player environments seems to have already been explored in [1]
> > ​​could the authors further elaborate on the connection between these two results
>
> All the methods are game-theoretic equilibrium-based rating schemes. In terms of differences in properties: Nash average is clone invariant, two-player zero-sum only, Nash equilibrium solution concept, and max-entropy equilibrium selection. Payoff ratings ([1]) are a *non-clone invariant*, n-player general-sum, correlated equilibrium solution concept, and use max-entropy equilibrium selection. Deviation ratings (this work) are clone invariant, n-player general-sum, coarse correlated equilibrium solution concept, and use min-max deviation rating equilibrium selection.
>
> Therefore deviation ratings combine the clone invariance of Nash average with the generality of payoff ratings. Achieving clone-invariance in n-player general-sum games was no easy task. We worked to achieve this property for a year. We are not aware of any other method that achieves this property.
>
> > since the rating is assessed under a correlated equilibrium, it would it would necessarily be negative (or <= epsilon) in the case of approximate CCE), which seems uninformative
> > For example, the rating scheme in [1] appears more intuitive.
>
> Concretely:
>
> NA: r_1(a’_1) = SUM_a_2 σ^*_2(a_2) G_p(a’_1, a_2)
>
> PR: r_p(a’_p) = SUM_a_-p σ^*_2(a_-p|a’_p) G_p(a’_p, a_-p)
>
> DR: r_p(a’_p) = SUM_a σ^*_2(a’_p, a_-p) [G_p(a’_p, a_-p) - G_p(a)]
>
> You are correct that the ratings will necessarily be nonpositive. They are made nonpositive by the SUM_a G_p(a) term in the definition. Note that this term is a constant over all player strategies (it does not feature a’_p). Therefore the offset is arbitrary and we could drop this term and all properties would still hold. Dropping the term would make the definition look similar to [1] (ignoring equilibrium selection differences). So nonpositive numbers in themselves are not uninformative.
>
> The advantage of the definition is that it communicates the ceiling of each strategies’ performance. Zero is the best rating a strategy can have in any game. Zero means the strategy is as good as an equilibrium distribution over many strategies and is perhaps dominant. In many games the best strategy has a negative rating, indicating that it is worse than an equilibrium distribution over strategies.
>
> > the authors state that they aim to minimize deviation gains, but it’s not intuitive why one would seek to minimize the so-called gains of a strategy
>
> The deviation gains are the change in payoff when deviating away from the equilibrium to a pure strategy. Therefore, in equilibrium, this will always be nonpositive. We recursively minimise the maximum deviation gain over all remaining strategies. This finds an equilibrium that is as strict as possible. As strictness increases, the payoffs under equilibrium (SUM_a G_p(a)) tend to increase, so deviating away from the equilibrium becomes even less favourable, and therefore deviation gains become more negative. So while deviation gains are being minimised the strictness of the equilibrium that the ratings are derived from is increasing which is a natural objective to optimise for. Indeed if we were to drop the SUM_a G_p(a) offset from the final rating (without any consequence to the algorithm), we would see the rating values would increase under such an optimization scheme.
>
> Another perspective is that it is difficult to minimise the deviation gain of a strong strategy. In the most extreme case, a dominant strategy’s deviation gain cannot be minimized below zero at all. So strong strategies counter the minimization scheme and remain highly rated.
>
> > Line 88: Since the concepts of WSCE and CE…
>
> We decided to keep this as we want to emphasise the fundamental nature of the deviation gains in the definitions of equilibrium solution concepts. Furthermore, the approximate discussion is relevant because the deviation gains are closely related to the approximation parameter.
>
> > Line 197: How are “deviation gain statistics” defined?
>
> We are simply referring to the “deviation gains”. We have dropped the term “statistics” from the text and agree it is causing confusion.
>
> > Line 225: Strategies are evaluated based on a correlated equilibrium computed from the deviation gains of players’ "uncovered" strategies. Intuitively, some properties may fail here, as they would be inherently dependent on the notion of correlated equilibrium.
>
> Section 4.2 lists properties that are satisfied by the proposed deviation ratings as computed by Algorithm 1, which implicitly computes a coarse correlated equilibrium. We would be happy to explain further if there is a specific question?

---

> > ### Author Response · Authors · 2024-12-02
> >
> > Thank you for your review. We hope you have had a chance to view our rebuttal?

---

> ### Author Response · Authors · 2024-11-25
>
> We hope our answers have helped with your concerns. If you have any other questions we would be happy to answer them while the communication channel is still open over the next 24 hours. Thank you for your time reviewing.

---

### Meta-Review · Area_Chair_Mzdd · 2024-12-20

**Metareview:**

This paper aims at ranking (or evaluating) the actions of a N-player game.

The game-theoretic motivation is highly unclear to me and the definition also not clear: what is $\sigma^*$ ? any CCE or one in particular. From the pseudo-code, it seems that it is the output of some computations, but not clear which ones.

Then the authors should prove more rigorously that cloning a strategy does not change the output of the algorithm; I am not 100% convinced about this.

But all in all, my main concerns are on the contributions of this paper that I found very slim. Maybe the previous rankings mentioned were not that great, but honestly, those discussed in the paper are rather obviously wrong (who thinks that the uniform value is a good idea for a game ?).

I am seriously concerned by the motivations, by the theoretical, conceptual and technical contributions of this paper. I do not believe it reaches the ICLR bar

**Additional Comments On Reviewer Discussion:**

Reviewers were not particularly excited by the paper, so I read it myself and my conclusions are listed above.

---

### Decision · Program_Chairs · 2025-01-22

Reject